# Flow2GAN: Hybrid Flow Matching and GAN with Multi-Resolution Network for Few-step High-Fidelity Audio Generation

**Zengwei Yao, Wei Kang, Han Zhu, Liyong Guo, Lingxuan Ye, Fangjun Kuang,**
**Weiji Zhuang, Zhaoqing Li, Zhifeng Han, Long Lin, Daniel Povey**
Xiaomi Corp., Beijing, China
dpovey@xiaomi.com

## Abstract

Existing dominant methods for audio generation include Generative Adversarial Networks (GANs) and diffusion-based methods like Flow Matching. GANs suffer from slow convergence during training, while diffusion methods require multi-step inference that introduces considerable computational overhead. In this work, we introduce Flow2GAN, a two-stage framework that combines Flow Matching training for learning generative capabilities with GAN fine-tuning for efficient few-step inference. Specifically, given audio's unique properties, we first improve Flow Matching for audio modeling through: 1) reformulating the objective as endpoint estimation, avoiding velocity estimation difficulties when involving empty regions; 2) applying spectral energy-based loss scaling to emphasize perceptually salient quieter regions. Building on these Flow Matching adaptations, we demonstrate that a further stage of lightweight GAN fine-tuning enables us to obtain few-step (e.g., 1/2/4 steps) generators that produce high-quality audio. In addition, we develop a multi-branch network architecture that processes Fourier coefficients at different time-frequency resolutions, which improves the modeling capabilities compared to prior single-resolution designs. Experimental results indicate that our Flow2GAN delivers high-fidelity audio generation from Mel-spectrograms or discrete audio tokens, achieving highly favorable quality-efficiency trade-offs compared to existing state-of-the-art GAN-based and Flow Matching-based methods. Online demo samples are available at `https://flow2gan.github.io`, and the source code is released at `https://github.com/k2-fsa/Flow2GAN`.

## 1 Introduction

Audio generation, also known as neural vocoding, aims to reconstruct high-resolution waveforms from compressed representations such as Mel-spectrograms and discrete audio tokens. It serves as a crucial component in various applications, including text-to-speech (TTS) (Shen et al., 2018; Kim et al., 2021; Zhu et al., 2025) and music synthesis (Engel et al., 2017; Donahue et al., 2018; Engel et al., 2019), directly impacting the perceptual experience of end users.

Deep generative models have demonstrated remarkable progress in this task. Generative Adversarial Networks (GANs) (Goodfellow et al., 2014; Arjovsky et al., 2017) are the dominant approach (Kumar et al., 2019; Kong et al., 2020a; Lee et al., 2022; Siuzdak, 2023), leveraging sophisticated discriminators (Kumar et al., 2019; Kong et al., 2020a; Jang et al., 2021; Gu et al., 2024) to capture multi-grained audio details and enabling high-fidelity results with efficient one-step inference. However, adversarial optimization in GANs often leads to slow convergence and the risk of mode collapse (Thanh-Tung & Tran, 2020). Another line of research explores the use of diffusion models (Sohl-Dickstein et al., 2015; Ho et al., 2020; Song et al., 2020) for audio generation (Kong et al., 2020b; Chen et al., 2020; Lee et al., 2021; Nguyen et al., 2024), offering stable training and strong generative performance. Recent diffusion-based works that integrate multi-band waveform processing (San Roman et al., 2023; Liu et al., 2024) further boost diffusion on audio data, enabling them to generate high-quality audio. Nevertheless, the strong results of diffusion models come at the cost of multi-step sampling, resulting in high computational demands.

In this work, we propose Flow2GAN, which follows a two-stage training strategy that combines the strengths of both GAN and diffusion paradigms. In the first stage, we train the model with a Flow Matching objective (Lipman et al., 2022; Liu et al., 2022a), a form of diffusion model (Kingma & Gao, 2023), to learn robust generative capabilities with stable training. In the second stage, we construct few-step generators from the trained model and utilize GAN fine-tuning for finer-grained generation through adversarial learning. To be specific, considering audio's distinctive properties, we improve Flow Matching for audio modeling by: 1) reformulating the training objective as direct endpoint prediction, circumventing the challenges of velocity estimation when involving empty regions; 2) incorporating spectral energy-based loss weighting to emphasize perceptually important low-energy regions. Building upon the adapted Flow Matching, the trained model is able to generate clearly improved audio with just 2 sampling steps (Figure 2). This provides a strong foundation for the subsequent GAN fine-tuning stage, which effectively guides the initialized few-step models toward further refined audio quality. Moreover, we design a multi-branch ConvNeXt-based (Liu et al., 2022b) network structure that operates on multi-resolution Fourier coefficients, serving as a powerful backbone with enhanced modeling capabilities compared to the previous single-branch version. Our experiments show that Flow2GAN generates high-quality audio across both Mel-spectrogram and audio token conditioning, while providing highly favorable quality-efficiency trade-offs compared to state-of-the-art GAN-based and Flow Matching-based methods. Ablation studies further validate the effectiveness of each proposed component.

## 2  RELATED WORK

**GAN-based audio generation.** GAN-based models consist of two networks: a generator that produces audio waveforms and a discriminator that classifies whether the generated audio is real or synthetic. The key to these methods is designing sophisticated discriminators to capture audio features from different perspectives, thereby guiding the generator to produce more fine-grained and realistic audio through adversarial learning. MelGAN (Kumar et al., 2019) employs a Multi-Scale Discriminator (MSD) that extracts features from waveforms at various downsampled resolutions. HiFi-GAN (Kong et al., 2020a) extends this approach by incorporating a Multi-Period Discriminator (MPD) that learns complementary structures by analyzing different periodic patterns. Univnet (Jang et al., 2021) utilizes a Multi-Resolution Discriminator (MRD) that operates in the time-frequency domain across multiple spectral resolutions. BigVGAN (Lee et al., 2022) introduces the Snake activation function to provide periodic inductive bias and incorporates low-pass filters for anti-aliasing. Unlike previous works that rely on transposed convolutions for upsampling, Vocos (Siuzdak, 2023) models spectral coefficients using ConvNeXt model (Liu et al., 2022b) and realizes upsampling to waveforms through the Inverse Short-Time Fourier Transform (ISTFT). Our network structure follows Vocos by using ConvNeXt to operate on spectral coefficients, but we extend it to multiple time-frequency resolutions for a more powerful backbone.

**Diffusion/Flow Matching-based audio generation.** Diffusion-based methods learn transition trajectories from noise distribution to data distribution, excelling at stable training and robust generation. DiffWave (Kong et al., 2020b) and WaveGrad (Chen et al., 2020) first introduce diffusion models to audio synthesis and get GAN-competitive results. PriorGrad (Lee et al., 2021) uses an adaptive prior derived from data statistics to structure the noise, thereby speeding up the diffusion sampling process. PeriodWave (Lee et al., 2024a) utilizes a multi-period Flow Matching estimator to learn different periodic features of the waveform. Multi-Band Diffusion (MBD) (San Roman et al., 2023) divides the audio into different frequency bands and processes each band with a distinct diffusion model. RFWave (Liu et al., 2024) also adopt this multi-band idea, employing Flow Matching to model each band but processing all bands in parallel with a shared model. These multi-band processing approaches employ frequency equalization to reduce the energy mismatch between the Gaussian distribution and audio data distribution in different frequency bands. While this technique is effective for high-frequency modeling, the equalization process relies on normalization statistics calculated on the dataset (San Roman et al., 2023) or accumulated during training (Liu et al., 2024), which poses a potential generalization risk when applied to distributions that differ significantly from those used to compute the original statistics. In this work, we improve Flow Matching for audio modeling by reformulating the objective to endpoint estimation and emphasizing loss in perceptually salient regions, and further combine a GAN fine-tuning stage for fast and finer-grained generation. Note that PriorGrad and RFWave also use energy-based loss scaling (Liu et al., 2024), but they operates at the per-frame level, whereas ours is based on both time and frequency dimensions, which is validated to outperform the per-frame version in our reformulated setting (Table 3).

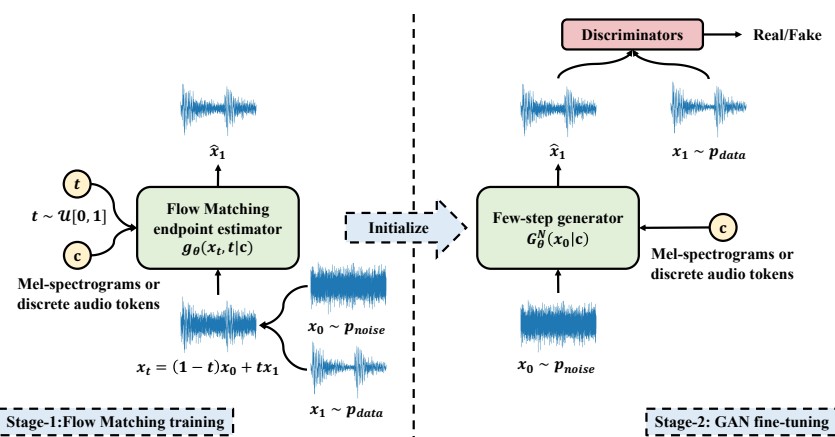

Figure 1: Overall framework of Flow2GAN.

**Accelerating Flow Matching models in audio generation.** A typical way to accelerate diffusion model inference is through distillation (Salimans & Ho, 2022; Meng et al., 2023; Song et al., 2023), using multi-step outputs to guide fewer-step outputs. WaveFM (Luo et al., 2025) introduces a tailored method of consistency distillation (Song et al., 2023) and can generate audio in 1 step without significant quality degradations. Another approach is to integrate GAN objectives with adversarial learning using real data (Huang et al., 2023; Lee et al., 2024b; Zheng & Yang, 2025). A recent work in audio generation is PeriodWave-Turbo (Lee et al., 2024b), which introduces GAN fine-tuning to Flow-Matching pre-trained models. Our Flow2GAN shares a similar idea, but we build upon our enhanced Flow Matching specialized for audio modeling, providing stronger pretrained weights to initialize the GAN generators. This results in significantly superior one-step generator after GAN fine-tuning compared to building upon standard Flow Matching (Table 3). Notably, PeriodWave-Turbo (Lee et al., 2024b) does not explore one-step results. Shortcut models (Frans et al., 2024) condition on both noise level and step size, enabling better few-step generation than consistency models (Song et al., 2023). We also investigate this strategy for few-step audio generation and demonstrate that our approach surpasses it.

## 3 PRELIMINARY: FLOW MATCHING

As a popular formulation of diffusion models (Sohl-Dickstein et al., 2015; Ho et al., 2020; Song et al., 2020; Rezende & Mohamed, 2015; Kingma & Gao, 2023), Flow Matching (Lipman et al., 2022; Liu et al., 2022a) addresses generative modeling by learning the velocity fields that transport the noise distribution $p_{\text{noise}} = \mathcal{N}(0, I)$ to the data distribution $p_{\text{data}}$. Given a noise point $x_0 \sim p_{\text{noise}}$, a data point $x_1 \sim p_{\text{data}}$, and $t \sim \mathcal{U}[0, 1]$, a straight flow path can be defined by linear interpolation: $x_t = (1 - t)x_0 + tx_1$, with corresponding velocity $v_t = x_1 - x_0$. The same intermediate point $x_t$ can arise from different pairs of $(x_0, x_1)$, each yielding a distinct velocity $v_t$. Flow Matching thus trains a network $f_\theta(x_t, t)$ to estimate the marginal velocity field (Lipman et al., 2022), that is, the expected velocity conditioned on $x_t$: $v(x_t, t) = \mathbb{E}_{x_0, x_1}[v_t | x_t]$. In practice, this is equivalently optimized in a simpler way by fitting the empirical velocity $v_t$ obtained from randomly sampled pairs $(x_0, x_1)$ during training (Lipman et al., 2022):

$$\mathcal{L}_{\text{FM}} = \mathbb{E}_{t, x_0, x_1} \left[ \| f_\theta(x_t, t) - v_t \|^2 \right]. \tag{1}$$

In inference, new data can be sampled by solving the ordinary differential equation (ODE) using the learned flow model: $\frac{dx_t}{dt} = f_\theta(x_t, t)$. This process is typically approximated by a numerical solver, such as Euler method, over discrete time steps:

$$x_{t_{i+1}} = x_{t_i} + (t_{i+1} - t_i)f_\theta(x_{t_i}, t_i). \tag{2}$$

## 4 METHOD

As illustrated in Figure 1, Flow2GAN follows a two-stage training strategy: the model is first trained with improved Flow Matching tailored for audio modeling (Section 4.1); then few-step generators are built with the trained model and guided by GAN fine-tuning toward fine-grained generation (Section 4.2). The model utilizes a multi-branch architecture that processes Fourier coefficients at different time-frequency resolutions (Section 4.3).

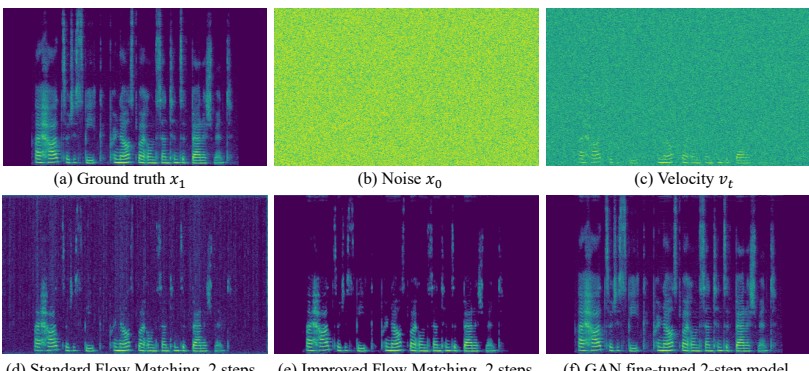

(a) Ground truth $x_1$       (b) Noise $x_0$       (c) Velocity $v_t$

(d) Standard Flow Matching, 2 steps    (e) Improved Flow Matching, 2 steps    (f) GAN fine-tuned 2-step model

Figure 2: Example illustration of audio generation. Flow Matching estimates velocity $v_t = x_1 - x_0$. The improved Flow Matching in Flow2GAN estimates endpoint $x_1$, yielding cleaner result in silent regions with 2 sampling steps. GAN fine-tuning further enhances the result by filling in details.

## 4.1 IMPROVED FLOW MATCHING FOR AUDIO MODELING

When applying Flow Matching to conditional audio generation, the velocity field is parameterized as $f_\theta(x_t, t|\mathbf{c})$, where $\mathbf{c}$ denotes the compressed acoustic representations, such as Mel-spectrograms or discrete audio tokens. However, the unique characteristics of audio data present two key challenges for Flow Matching in this context: 1) **Velocity estimation difficulty.** Audio signals often contain silent segments or frequency bands with zero energy (Figure 2 (a)). In these regions, the model must accurately estimate $v_t$ to precisely cancel out the noise in $x_0$ and recover the empty target $x_1$. This increases the learning difficulty as the model is required not only to capture the differences $v_t = x_1 - x_0$ in the active regions of the audio signal, but also to approximate $-x_0$ in these empty regions (Figure 2 (c)). 2) **Loss-perception mismatch.** The Flow Matching loss (Equation 1) uses mean squared error (MSE) criterion, treating the prediction errors uniformly across all signal regions. However, this does not align well with auditory perception principles, where errors in quieter spectral regions tend to be more perceptually noticeable than equivalent errors in louder regions.

**Reformulation to endpoint estimation.** To circumvent the difficulty of predicting the velocity $v_t$, we reformulate the Flow Matching problem so that the network predicts the endpoint $x_1$ instead: $\hat{x}_1 = g_\theta(x_t, t|\mathbf{c})$. Specifically, as $v_t = \frac{x_1 - x_t}{1 - t}$, the loss in Equation 1 can be rewritten as:

$$\mathcal{L}_{\text{FM}} = \mathbb{E}_{t, x_0, x_1}\left[\left\|\frac{g_\theta(x_t, t|\mathbf{c}) - x_t}{1 - t} - \frac{x_1 - x_t}{1 - t}\right\|^2\right] = \mathbb{E}_{t, x_0, x_1}\left[\frac{\|g_\theta(x_t, t|\mathbf{c}) - x_1\|^2}{(1 - t)^2}\right]. \quad (3)$$

This reformulation can be interpreted as training the network to reconstruct the clean audio $x_1$ from its noisy version $x_t$ at varying noise levels $(1 - t)$. This approach provides a more consistent and stable learning objective tailored to audio data, allowing the network to focus on capturing audio-relevant patterns regardless of whether the region is active or empty. Empirically, we found that removing the weighting factor $\frac{1}{(1-t)^2}$ in Equation 3 yields slight improvements at both the Flow-Matching stage (2 steps) and after GAN fine-tuning. The reason might be that without this weighting, the model can better emphasize small $t$ values, which aligns well with our setting. For simplicity, we therefore omit this factor, yielding:

$$\mathcal{L}'_{\text{FM}} = \mathbb{E}_{t, x_0, x_1}\left[\|g_\theta(x_t, t|\mathbf{c}) - x_1\|^2\right]. \quad (4)$$

The sampling process in Equation 2 is consequently modified as follows:

$$x_{t_{i+1}} = x_{t_i} + (t_{i+1} - t_i)\frac{g_\theta(x_{t_i}, t_i|\mathbf{c}) - x_{t_i}}{1 - t_i}. \quad (5)$$

**Spectral energy-adaptive loss scaling.** To address the loss-perception mismatch, we scale the Flow Matching loss inversely proportional to the energy of the reference spectrogram, thereby encouraging the model to emphasize the perceptually salient quieter regions. Specifically, the prediction error is converted into frequency domain: $\mathcal{S}(g_\theta(x_t, t|\mathbf{c}) - x_1)$, where $\mathcal{S}(x) = \text{LinFB}(|\text{STFT}(x)|^2)$ denotes the power STFT followed by a linear filterbank transformation for energy smoothing, and is differentiable. The obtained result is then scaled element-wise inversely by the square root of the

reference spectrogram's energy, $\frac{1}{\sqrt{\mathcal{S}(x_1)+\epsilon}}$, with $\epsilon = 1e-7$. Therefore, the loss in Equation 4 is redefined as:

$$\mathcal{L}''_{\text{FM}} = \mathbb{E}_{t,x_0,x_1}\left[\sum_{i,j}\left(\frac{\mathcal{S}(g_\theta(x_t,t|\mathbf{c})-x_1)}{\sqrt{\mathcal{S}(x_1)+\epsilon}}\right)_{i,j}\right]. \tag{6}$$

We clamp the loss scale $\frac{1}{\sqrt{\mathcal{S}(x_1)+\epsilon}}$ between 0.01 and 100 for training stability. See Appendix Table 10 for detailed configurations of $\mathcal{S}(x)$. Note that unlike the per-frame energy-based scaling in previous works (Lee et al., 2021; Liu et al., 2024), our scaling strategy is more comprehensive as it also accounts for differences along the frequency dimension.

## 4.2 GAN FINE-TUNING FOR REFINEMENT AND FEW-STEP INFERENCE

Although the adaptations described in Section 4.1 improve the audio generation quality of the Flow Matching model with 2 sampling steps (see Figure 2 (d) to (e) and Table 3), the inherent complexity of audio generation still leaves room for further refinement of details, especially in the high-frequency range. Moreover, even though we can obtain higher quality results through multi-step sampling (see Table 9 in Appendix), this would require considerable computational overhead. To this end, we employ a GAN fine-tuning strategy for two purposes: enhancing the fine-grained audio details by leveraging the existing carefully designed discriminators (Kong et al., 2020a; Jang et al., 2021) that capture diverse audio patterns from multiple perspectives, and enabling high-fidelity generation with few-step (even only one-step) inference.

We construct the few-step GAN generators, referred to as $G^N_\theta(x_0|\mathbf{c})$, by forwarding the trained Flow Matching model through $N$ steps following Equation 5. Note that generators with different steps are trained and deployed as separate models rather than a single unified model. Specifically, for $G^N_\theta(x_0|\mathbf{c})$ with $N > 1$, gradients are backpropagated through both forward passes, enabling end-to-end optimization of the intermediate output from earlier steps. While this approach can extend to more steps, we focus on one-, two-, and four-step variants for computational efficiency. Following (Siuzdak, 2023), we adopt the multi-period discriminator (MPD) (Kong et al., 2020a) and the multi-resolution discriminator (MRD) (Jang et al., 2021) for adversarial training. The generators are fine-tuned using a combination of HingeGAN adversarial loss (Lim & Ye, 2017), $L1$ feature matching loss, and multi-scale $L1$ Mel-spectrogram reconstruction loss (Kumar et al., 2023).

We found that a fast stage of GAN fine-tuning is sufficient to yield a rapid and substantial improvement in audio quality, while additional fine-tuning can provide further gains (see Table 4). Figure 2 (e) to (f) shows that the fine-grained details are recovered after GAN fine-tuning. Intuitively, this effect arises because the pre-training stage with the improved Flow Matching objective already equips the model with strong generative modeling capabilities, while subsequent fine-tuning efficiently guides the model toward high-fidelity output by iteratively refining details through adversarial learning with real audio samples. As shown in Table 3, it is worth noting that when using standard Flow Matching for pre-training like (Lee et al., 2024b), the one-step model achieves significantly lower performance after GAN fine-tuning, demonstrating the superiority of our improved Flow Matching objective. This strategy combines the strengths of both paradigms - improved Flow Matching for robust generative capability learning, and GAN for detail enhancement - ultimately enabling high-fidelity audio generation with fast inference. Notably, compared to pure GAN training, it also requires significantly lower training cost.

## 4.3 MULTI-RESOLUTION NETWORK STRUCTURE

Inspired by Vocos (Siuzdak, 2023), our model processes Fourier coefficients rather than waveform samples for saving computational cost and memory usage. While Vocos operates at a single time-frequency resolution, we extend to multi-resolution modeling to better capture audio complexity, providing a more powerful backbone for generative learning.

As shown in Figure 3, our model comprises three branches, each processing Fourier coefficients at different resolutions. Specifically, in each branch, the input signal is first transformed via STFT to obtain complex Fourier coefficients, whose real and imaginary components are concatenated along the feature dimension and fed into a ConvNeXT model (Liu et al., 2022b) to produce output complex coefficients. These are then converted back to waveform domain via Inverse STFT (ISTFT). The

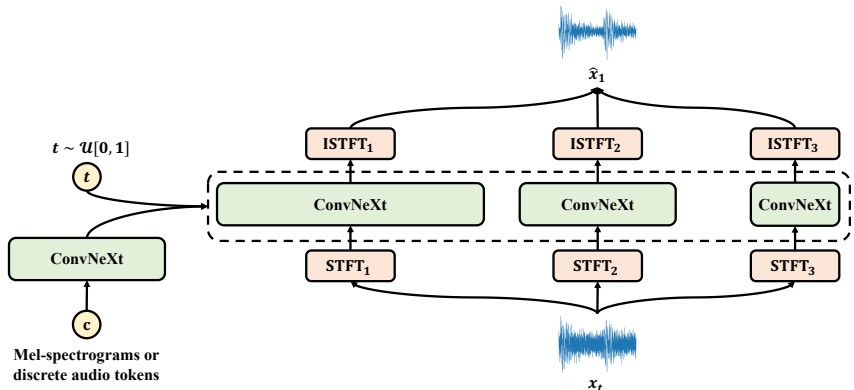

Figure 3: Multi-resolution network structure. Our model processes spectral coefficients from STFT at three different time-frequency resolutions, using larger embedding dimensions for shorter sequences. A ConvNeXt-based encoder processes the conditional compressed representation.

final output is obtained by summing all branch outputs. Since both STFT and ISTFT operations are differentiable, the entire model is optimized end-to-end. Additionally, following (Lee et al., 2024a), we incorporate a ConvNeXt-based condition encoder to extract deeper features from input Mel-spectrograms or codec token embeddings, which serve as shared generation conditions across all branches. Note that for Flow Matching inference, this condition encoder requires only one forward pass, with output features reused across multiple sampling steps. To better balance performance and efficiency, we use larger embedding dimensions for the low-frame-rate branch, and smaller embedding dimensions for the two high-frame-rate branches. Detailed model configurations are provided in Appendix 10.

## 5 EXPERIMENTS

To evaluate the effectiveness of our Flow2GAN, we conduct experiments conditioned on Mel-spectrograms and discrete audio tokens from Encodec (Défossez et al., 2022) (Section 5.2). We also perform ablation studies with Mel-spectrogram condition to investigate the effect of each proposed technique (Section 5.3) and compare inference speed with other models (Section 5.4). Finally, we evaluate performance when serving as a TTS vocoder (Section 5.5).

### 5.1 EXPERIMENTAL SETUP

**Datasets and evaluation metrics.** For Mel-spectrogram conditioning, we train models on the LibriTTS dataset (Zen et al., 2019), which contains 585 hours of English speech at 24kHz sampling rate. We report results on the test set for model comparisons and on the dev set for ablation studies. For objective evaluation, we use Perceptual Evaluation of Speech Quality (PESQ) (Rix et al., 2001), ViSQOL (Chinen et al., 2020), F1 score for voiced/unvoiced classification (V/UV F1), periodicity error (Periodicity). We also include Fréchet Speech Distance (FSD) (Le et al., 2023), which measures the similarity between distributions of generated and real samples in a feature space from a Wav2Vec2 encoder (Baevski et al., 2020). For subjective evaluation, we use Similarity MOS (SMOS) between generated and reference audios, with scores from 1 (not at all similar) to 5 (extremely similar), and also 5-scale Mean Opinion Score (MOS) to evaluate naturalness with scores from 1 (completely unnatural) to 5 (completely natural). We compare against the following state-of-the-art models: BigVGAN (Lee et al., 2022), Vocos (Siuzdak, 2023), RFWave (Liu et al., 2024), PeriodWave-Turbo (Lee et al., 2024b), and WaveFM (Luo et al., 2025). In ablation studies, we focus on PESQ and ViSQOL. We use log-Mel spectrograms when comparing to other models and Mel spectrograms in ablation studies, with their results comparison presented in Appendix Table 11.

For audio token conditioning, following (San Roman et al., 2023; Liu et al., 2024), we train on a set of universal audio datasets including: Common Voice 7.0 (Ardila et al., 2019), clean speech data from DNS Challenge 4 (Dubey et al., 2024) (speech), MTG-Jamendo (Bogdanov et al., 2019) (music), and AudioSet (Gemmeke et al., 2017) and FSD50K (Fonseca et al., 2021) (sound events). All audio files are resampled to 24kHz. Following (Liu et al., 2024), we evaluate on a unified test dataset[1], which contains 900 test samples involving speech, vocals, and sound effect. For the

---

[1]https://github.com/bfs18/rfwave

universal audios, we report PESQ, ViSQOL and FSD as objective metrics, as well as SMOS and MOS as subjective metric. In this setting, we compare Flow2GAN against Encodec (Défossez et al., 2022), MBD (San Roman et al., 2023), RFWave (Liu et al., 2024), and PeriodWave-Turbo (Lee et al., 2024b).

To evaluate Flow2GAN as a vocoder in Mel-based TTS systems, we combine it with F5-TTS Base (Chen et al., 2025), a recent zero-shot TTS model, and compare against various state-of-the-art vocoders. Note that F5-TTS provides two official checkpoints corresponding to librosa and torchaudio mel-spectrogram implementations[2]. We evaluate zero-shot TTS performance on LibriSpeech-PC (Meister et al., 2023) test-clean subset. For intelligibility, we report WER between the synthesized speech transcription and the input text. Speaker similarity is measured using cosine similarity between WavLM-based (Chen et al., 2022) ECAPA-TDNN (Desplanques et al., 2020) embeddings of the prompt and synthesized speech (denoted as SIM-o (Le et al., 2023)). Naturalness is evaluated using network-based MOS (UTMOS) (Saeki et al., 2022).

**Implementation details.** Our Flow2GAN is built with ConvNeXt (Liu et al., 2022b) layers, but we replace the normalization with BiasNorm (Yao et al., 2024) and use PreLU activation (He et al., 2015). Detailed model configurations are provided in Appendix Table 10. In GAN fine-tuning stage, following (Kumar et al., 2023), we use multi-scale Mel-spectrogram reconstruction loss with window lengths of $\{32, 64, 128, 256, 512, 1024, 2048\}$, hop lengths set to window length / 4, and Mel bins of $\{5, 10, 20, 40, 80, 160, 320\}$. For Encodec audio token conditioning, during training we randomly choose a bandwidth from $\{12.0, 6.0, 3.0, 1.5\}$ kbps. At inference, the unified model can support all these bandwidths. We train our models with ScaledAdam optimizer (Yao et al., 2024), which offers faster convergence. Mel-spectrogram conditioned models are trained for 92k Flow Matching iterations and 110k GAN fine-tuning iterations on 2 Nvidia H20 GPUs. Audio token conditioned models are trained for 180k Flow Matching iterations (8 H20 GPUs) and 190k GAN fine-tuning iterations (2 H20 GPUs). For the state-of-the-art models for comparison, we use their released checkpoints and recommended inference configurations.

### 5.2 COMPARISON TO STATE-OF-THE-ART METHODS

**Mel-spectrogram conditioning.** Table 1 compares our Flow2GAN models with other state-of-the-art models with Mel-spectrogram conditioning. The one-step Flow2GAN outperforms Vocos (Siuzdak, 2023), RFWave (Liu et al., 2024), and WaveFM (Luo et al., 2025) across most metrics, except PESQ. The two-step and four-step variants outperform all other models except for FSD, SMOS, and MOS. The four-step version achieves the best results in PESQ, ViSQOL, V/UV F1, and Periodicity. As BigVGAN-v2 (Lee et al., 2022) is trained on a larger dataset [3], achieving results close to it demonstrates the effectiveness of our models.

Table 1: Comparison to state-of-the-art methods on LibriTTS test set with Mel-spectrogram conditioning. *BigVGAN-v2 is trained on a large-scale dataset.

| Model | Params (M) | PESQ↑ | ViSQOL↑ | V/UV F1↑ | Periodicity↓ | FSD↓ | SMOS↑ | MOS↑ |
|---|---|---|---|---|---|---|---|---|
| BigVGAN | 112.4 | 4.241 | 4.964 | 0.969 | 0.071 | 0.022 | 4.47 ± 0.10 | 4.33 ± 0.18 |
| BigVGAN-v2* | 112.4 | 4.379 | 4.971 | 0.978 | 0.055 | **0.014** | **4.65 ± 0.11** | **4.59 ± 0.10** |
| Vocos | 13.5 | 3.618 | 4.898 | 0.951 | 0.105 | 0.042 | 4.10 ± 0.17 | 4.38 ± 0.16 |
| RFWave (10 steps) | 18.1 | 4.220 | 4.772 | 0.957 | 0.098 | 0.412 | 4.24 ± 0.16 | 4.29 ± 0.13 |
| PeriodWave-Turbo (4 steps) | 70.2 | 4.434 | 4.965 | 0.958 | 0.096 | 0.020 | 4.20 ± 0.17 | 4.38 ± 0.17 |
| WaveFM (1 step) | 19.5 | 3.540 | 4.894 | 0.943 | 0.124 | 0.098 | 3.72 ± 0.18 | 3.76 ± 0.18 |
| Flow2GAN, 1-step (**ours**) | 78.9 | 4.189 | 4.957 | 0.975 | 0.063 | 0.028 | 4.44 ± 0.14 | 4.39 ± 0.15 |
| Flow2GAN, 2-step (**ours**) | 78.9 | 4.440 | 4.979 | 0.983 | 0.044 | 0.023 | 4.53 ± 0.13 | 4.56 ± 0.11 |
| Flow2GAN, 4-step (**ours**) | 78.9 | **4.484** | **4.986** | **0.985** | **0.037** | 0.016 | 4.60 ± 0.14 | 4.58 ± 0.14 |

**Encodec audio token conditioning.** Table 2 presents a comparison of Flow2GAN models with other state-of-the-art models conditioned on EnCodec audio tokens. At lower bandwidths of 1.5 and 3.0 kbps, all Flow2GAN variants outperform other models across most objective metrics, except that the one-step and two-step models underperform PeriodWave-Turbo (Lee et al., 2024b) in ViSQOL at 3.0 kbps. At higher bandwidths of 6.0 and 12.0 kbps, although PeriodWave-Turbo achieves the best results in PESQ and ViSQOL, all Flow2GAN variants outperform other models across all objective metrics, with the four-step Flow2GAN achieving notably better FSD results than PeriodWave-Turbo. With increasing bandwidth, all Flow2GAN models can consistently achieve better SMOS and MOS.

[2] https://github.com/SWivid/F5-TTS

[3] https://github.com/NVIDIA/BigVGAN

Table 2: Comparison to state-of-the-art methods on universal evaluation dataset with Encodec audio token conditioning.

| Bandwidth (kbps) | Model | Params (M) | PESQ↑ | ViSQOL↑ | FSD ↓ | SMOS↑ | MOS↑ |
|---|---|---|---|---|---|---|---|
| 1.5 | EnCodec | 56.0 | 1.368 | 3.409 | 1.996 | $1.69 \pm 0.14$ | $2.09 \pm 0.27$ |
| | MBD | 411.0 | 1.457 | 3.030 | 7.734 | $2.67 \pm 0.18$ | $2.83 \pm 0.28$ |
| | RFWave (10-step) | 18.1 | 1.600 | 3.272 | 2.986 | $2.87 \pm 0.23$ | $2.91 \pm 0.23$ |
| | PeriodWave-Turbo (4-step) | 31.3 | 1.260 | 3.308 | 4.055 | $1.55 \pm 0.16$ | $1.47 \pm 0.19$ |
| | Flow2GAN, 1-step (**ours**) | 78.9 | 1.739 | 3.582 | 1.210 | $2.43 \pm 0.20$ | $2.83 \pm 0.21$ |
| | Flow2GAN, 2-step (**ours**) | 78.9 | 1.803 | 3.609 | 1.152 | $3.04 \pm 0.20$ | $\mathbf{3.86 \pm 0.14}$ |
| | Flow2GAN, 4-step (**ours**) | 78.9 | **1.925** | **3.662** | **1.069** | $\mathbf{3.17 \pm 0.19}$ | $3.40 \pm 0.18$ |
| 3.0 | EnCodec | 56.0 | 1.745 | 3.867 | 1.380 | $3.11 \pm 0.26$ | $3.31 \pm 0.23$ |
| | MBD | 411.0 | 1.958 | 3.523 | 6.133 | $3.36 \pm 0.21$ | $4.00 \pm 0.24$ |
| | RFWave (10-step) | 18.1 | 2.124 | 3.784 | 2.622 | $3.41 \pm 0.21$ | $3.86 \pm 0.23$ |
| | PeriodWave-Turbo (4-step) | 31.3 | 2.160 | 4.058 | 1.018 | $3.04 \pm 0.17$ | $3.16 \pm 0.23$ |
| | Flow2GAN, 1-step (**ours**) | 78.9 | 2.353 | 4.026 | 0.867 | $3.94 \pm 0.14$ | $4.00 \pm 0.19$ |
| | Flow2GAN, 2-step (**ours**) | 78.9 | 2.442 | 4.049 | 0.843 | $\mathbf{4.19 \pm 0.15}$ | $4.07 \pm 0.24$ |
| | Flow2GAN, 4-step (**ours**) | 78.9 | **2.550** | **4.091** | **0.804** | $4.03 \pm 0.16$ | $\mathbf{4.08 \pm 0.22}$ |
| 6.0 | EnCodec | 56.0 | 2.259 | 4.156 | 1.239 | $3.77 \pm 0.19$ | $3.47 \pm 0.21$ |
| | MBD | 411.0 | 2.094 | 3.690 | 6.177 | $3.15 \pm 0.19$ | $4.47 \pm 0.21$ |
| | RFWave (10-step) | 18.1 | 2.663 | 4.120 | 2.574 | $3.68 \pm 0.22$ | $4.00 \pm 0.22$ |
| | PeriodWave-Turbo (4-step) | 31.3 | **3.229** | **4.424** | 0.712 | $4.00 \pm 0.17$ | $4.40 \pm 0.21$ |
| | Flow2GAN, 1-step (**ours**) | 78.9 | 2.904 | 4.300 | 0.696 | $\mathbf{4.46 \pm 0.16}$ | $\mathbf{4.42 \pm 0.22}$ |
| | Flow2GAN, 2-step (**ours**) | 78.9 | 2.983 | 4.319 | 0.733 | $4.38 \pm 0.14$ | $4.31 \pm 0.17$ |
| | Flow2GAN, 4-step (**ours**) | 78.9 | 3.089 | 4.351 | **0.678** | $4.19 \pm 0.12$ | $4.38 \pm 0.13$ |
| 12.0 | EnCodec | 56.0 | 2.776 | 4.347 | 1.204 | $3.76 \pm 0.15$ | $4.15 \pm 0.15$ |
| | MBD | 411.0 | - | - | - | - | - |
| | RFWave (10-step) | 18.1 | 3.117 | 4.348 | 2.152 | $3.75 \pm 0.18$ | $4.13 \pm 0.23$ |
| | PeriodWave-Turbo (4-step) | 31.3 | **3.579** | **4.544** | 0.776 | $4.32 \pm 0.16$ | $\mathbf{4.56 \pm 0.16}$ |
| | Flow2GAN, 1-step (**ours**) | 78.9 | 3.389 | 4.489 | 0.632 | $\mathbf{4.52 \pm 0.13}$ | $4.53 \pm 0.23$ |
| | Flow2GAN, 2-step (**ours**) | 78.9 | 3.457 | 4.505 | 0.600 | $4.39 \pm 0.14$ | $4.44 \pm 0.28$ |
| | Flow2GAN, 4-step (**ours**) | 78.9 | 3.538 | 4.531 | **0.557** | $4.22 \pm 0.19$ | $4.50 \pm 0.15$ |

## 5.3 ABLATION STUDIES

**Ablation of the improved Flow Matching.** Table 3 presents ablation results for the Flow Matching stage in Flow2GAN with Mel-spectrogram conditioning. Since we aim for few-step generation after GAN fine-tuning, we focus on the 2-step sampling results for the Flow Matching model. We don't compare one-step Flow Matching results as they tend to converge toward the dataset mean (Frans et al., 2024). The results indicate that reformulating the standard Flow Matching objective to endpoint prediction improves PESQ by 0.455 and also yields better results after GAN fine-tuning (from Equation 1 to Equation 4). Additionally, the proposed spectral energy-adaptive loss scaling (Equation 6) further improves PESQ and ViSQOL substantially and consistently in both stages. Per-frame energy-based scaling, as in (Lee et al., 2021; Liu et al., 2024), leads to clear performance drops in both stages, demonstrating the advantages of our energy-based scaling strategy across time and frequency dimensions. Maintaining the factor $\frac{1}{(1-t)^2}$ in Equation 3 yields worse results, as it emphasizes larger $t$ values, which is not suitable for our few-step goal. The consistent improvements across both stages confirm that our Flow Matching enhancements contribute to the final performance gains, distinguishing our approach from PeriodWave-Turbo (Lee et al., 2024b), which uses standard Flow Matching for pre-training.

Table 3: Ablation study of the improved Flow Matching on LibriTTS dev set with Mel-spectrogram conditioning.

| Method | Flow Matching trained (2 steps) | | GAN fine-tuned | | | |
|---|---|---|---|---|---|---|
| | | | 1-step | | 2-step | |
| | PESQ↑ | ViSQOL↑ | PESQ↑ | ViSQOL↑ | PESQ↑ | ViSQOL↑ |
| Standard Flow Matching | 2.351 | 3.691 | 3.730 | 4.853 | 4.257 | 4.933 |
| Predict $x_1$, w/o loss scaling | 2.806 | 3.691 | 4.173 | 4.912 | 4.332 | 4.937 |
| Predict $x_1$, w/ per-frame loss scaling | 3.140 | 3.667 | 4.201 | 4.910 | 4.388 | 4.942 |
| Predict $x_1$, w/ loss scaling, w/ loss factor $\frac{1}{(1-t)^2}$ | 3.226 | 3.726 | 4.195 | 4.932 | 4.430 | 4.962 |
| Predict $x_1$, w/ loss scaling (**final**) | **3.469** | **3.749** | **4.303** | **4.942** | **4.471** | **4.969** |

**Comparison to pure GAN training.** Table 4 demonstrates the effectiveness of our two-stage training paradigm. Pure GAN training converges slowly, whereas Flow2GAN achieves significantly better results with lower training costs by combining Flow Matching training with subsequent GAN

fine-tuning. Notably, even a fast GAN fine-tuning with 11k iterations yields high-quality generations for both the one-step and two-step generators, validating the benefits of this training strategy.

Table 4: Comparison to pure GAN training on LibriTTS dev set with Mel-spectrogram conditioning.

| Method | Training iterations | Training hours | PESQ↑ | ViSQOL↑ |
|---|---|---|---|---|
| Pure GAN | 110k | 26 | 3.587 | 4.823 |
| | 220k | 53 | 3.770 | 4.838 |
| | 330k | 78 | 3.838 | 4.861 |
| | 660k | 156 | **3.919** | **4.888** |
| Flow Matching trained (2 steps) | 92k | 50 | 3.469 | 3.749 |
| + GAN fine-tuned, 1-step | +11k | +2.6 | 4.195 | 4.887 |
| | +55k | +13 | 4.286 | 4.933 |
| | +110k | +26 | **4.303** | **4.942** |
| + GAN fine-tuned, 2-step | +11k | +3.2 | 4.375 | 4.917 |
| | +55k | +16 | 4.457 | 4.961 |
| | +110k | +32 | **4.471** | **4.969** |

**Comparison to Shortcut Models.** We also investigate Shortcut models (Frans et al., 2024) for few-step audio generation. Specifically, we allocate 75% of each batch to Flow Matching loss and 25% to self-consistency loss. See Algorithms 1, 2 in (Frans et al., 2024) for training and sampling details. As shown in Table 5, Shortcut models improve few-step results over standard Flow Matching in both PESQ and ViSQOL. Our Flow2GAN achieves further performance improvements, demonstrating clear advantages.

Table 5: Comparison to Shortcut models on LibriTTS dev set with Mel-spectrogram conditioning.

| Method | Steps | PESQ↑ | ViSQOL↑ |
|---|---|---|---|
| Standard Flow Matching | 1 | 1.132 | 1.801 |
| | 2 | 2.351 | 3.691 |
| | 4 | 3.220 | 4.173 |
| Shortcut Models | 1 | 3.181 | 4.426 |
| | 2 | 3.841 | 4.660 |
| | 4 | 4.122 | 4.781 |
| Flow2GAN | 1 | 4.303 | 4.942 |
| | 2 | 4.471 | **4.969** |
| | 4 | **4.498** | 4.967 |

**Ablation of the multi-resolution network.** As shown in Table 6, our multi-resolution network structure outperforms the single-resolution baseline (Siuzdak, 2023) with comparable parameters in the final two-step system, demonstrating the benefits of modeling multi-resolution spectral features. Removing the condition encoder in Flow2GAN causes a performance drop, indicating that learning higher-level conditional features is helpful.

Table 6: Ablation study of model structure on LibriTTS dev set with Mel-spectrogram conditioning.

| Method (2-step) | Params (M) | PESQ↑ | ViSQOL↑ |
|---|---|---|---|
| Single-resolution, double layers | 79.9 | 4.442 | 4.920 |
| Multi-resolution, w/o condition encoder | 62.4 | 4.417 | 4.966 |
| Multi-resolution (**final**) | 78.9 | **4.471** | **4.969** |

## 5.4 INFERENCE SPEED

In table 7, we compare Flow2GAN with state-of-the-art models under Mel-spectrogram conditioning in terms of inference speed, measured by the speed factor relative to real time (xRT). Experiments are conducted on an Intel Xeon Platinum 8457C CPU and an NVIDIA H20 GPU with batch size of 16 and segment length of 1 second. Except for Vocos, all versions of Flow2GAN achieve significantly faster inference than other models on both CPU and GPU, while even running faster than real time on CPU. This demonstrates that Flow2GAN excels not only in quality but also in efficiency.

Table 7: Model generation speed comparison with batch size of 16 and length of 1 second. BigVGAN-v2* use the specialized CUDA kernel.

| Model | Params (M) | xRT↑ | |
| --- | --- | --- | --- |
| | | CPU | GPU |
| BigVGAN | 112.4 | 0.214 | 69.8 |
| BigVGAN-v2 | 112.4 | 0.214 | 69.8 |
| BigVGAN-v2* | 112.4 | - | 121.9 |
| Vocos | 13.5 | **387.57** | **6440.80** |
| RFWave (10-step) | 18.1 | 0.37 | 158.8 |
| PeriodWave-Turbo (4-step) | 70.24 | 0.12 | 43.70 |
| WaveFM (1-step) | 19.5 | 0.64 | 226.31 |
| Flow2GAN, 1-step **(ours)** | 78.9 | 4.85 | 851.67 |
| Flow2GAN, 2-step **(ours)** | 78.9 | 2.46 | 449.26 |
| Flow2GAN, 4-step **(ours)** | 78.9 | 1.35 | 228.48 |

## 5.5 ZERO-SHOT TTS PERFORMANCE COMPARISON

As shown in Table 8, our 4-step Flow2GAN achieves the same SIM-o and slightly better UTMOS than PeriodWave-Turbo (Lee et al., 2024b) with significantly faster inference (Table 7), though with higher WER. Notably, BigVGAN-v2 performs worse than BigVGAN (Lee et al., 2022), suggesting potential overfitting to ground-truth Mel-spectrograms. We also find that adding small Gaussian noise $0.2 \times \text{rand}() \times \mathcal{N}(0, 1)$ to the conditioning log-Mel during GAN fine-tuning improves both WER and UTMOS, possibly because it makes the model more robust to the imperfect spectrograms from the TTS diffusion model.

Table 8: Zero-shot TTS results on LibriSpeech-PC test-clean with F5-TTS Base models (librosa/torchaudio Mels) and various vocoders. *BigVGAN-v2 is trained on a large-scale dataset. [†] denotes adding noise to log-Mel during GAN fine-tuning.

| Model | Params (M) | Mel-type | WER(%)↓ | SIM-o↑ | UTMOS↑ |
| --- | --- | --- | --- | --- | --- |
| BigVGAN | 112.4 | librosa | **1.78** | 0.67 | 4.05 |
| BigVGAN-v2* | 112.4 | | 1.82 | 0.66 | 3.70 |
| PeriodWave-Turbo (4 steps) | 70.2 | | 1.8 | **0.67** | 4.17 |
| Vocos | 13.5 | torchaudio | 1.91 | 0.65 | 3.90 |
| RFWave (10 steps) | 18.1 | | 1.87 | 0.66 | 3.63 |
| WaveFM (1 step) | 19.5 | | 2.01 | 0.65 | 3.15 |
| Flow2GAN, 1-step **(ours)** | 78.9 | | 1.94 | **0.67** | 3.75 |
| Flow2GAN, 2-step **(ours)** | 78.9 | | 1.91 | **0.67** | 4.11 |
| Flow2GAN, 4-step **(ours)** | 78.9 | | 1.89 | **0.67** | 4.18 |
| Flow2GAN, 1-step[†] **(ours)** | 78.9 | | 1.88 | **0.67** | 3.83 |
| Flow2GAN, 2-step[†] **(ours)** | 78.9 | | 1.88 | **0.67** | 4.21 |
| Flow2GAN, 4-step[†] **(ours)** | 78.9 | | 1.87 | **0.67** | **4.26** |

## 6 CONCLUSION

In this work, we propose Flow2GAN, a two-stage training framework. In the first stage, Flow Matching is used to learn the generative capability, while in the second stage, GAN fine-tuning refines the details, ultimately enabling few-step generation. Specifically, we improve Flow Matching by reformulating to endpoint estimation to mitigate the difficulties of velocity prediction in empty regions, and by introducing spectral energy–adaptive loss scaling to emphasize perceptually salient quieter regions. Based on this, we construct few-step generators from the trained model, and demonstrate that incorporating an additional GAN stage can efficiently enhance the results. Our backbone adopts a multi-branch network that processes spectral coefficients at different time–frequency resolutions, providing powerful modeling capabilities. Experimental results demonstrate that Flow2GAN achieves efficient high-quality audio generation within few steps (e.g., 1, 2, and 4 steps), delivering highly favorable quality-speed trade-offs in few-step generation compared to existing methods.

## REPRODUCIBILITY STATEMENT

The datasets used in our experiments are publicly available. Implementation details are provided in Section 5.1, and model configurations are described in Appendix Section A.3. To facilitate the reproduction of our results, the source code and pretrained checkpoints are publicly available at `https://github.com/k2-fsa/Flow2GAN`.

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

# A    APPENDIX

## A.1    USE OF LARGE LANGUAGE MODELS (LLMs)

In this work, we used LLM as a writing assistance tool to improve phrasing, grammar, and sentence clarity.

## A.2    MULTI-STEP SAMPLING RESULT

Table 9 presents the multi-step sampling results in terms of PESQ and ViSQOL for the Flow Matching stage in Flow2GAN. The results show that increasing the number of sampling steps improves generation quality.

Table 9: Multi-step results on LibriTTS dev set for the Flow Matching stage of Flow2GAN.

| Inference steps | PESQ↑ | ViSQOL↑ |
|---|---|---|
| 1 | 2.279 | 2.222 |
| 2 | 3.469 | 3.749 |
| 4 | 3.788 | 4.156 |
| 8 | 4.013 | 4.392 |
| 16 | 4.166 | 4.553 |

## A.3    MODEL CONFIGURATION

Table 10: Detailed model configurations of Flow2GAN with Mel-spectrogram and Encodec audio token conditioning.

| Configuration | Mel-spectrogram | Encodec audio token |
|---|---|---|
| Input condition | N-FFT=1024,Hop=256,N-Mel=100 | Latent-dim=128,Hop=320 |
| Time embedding dimension | 512 | 512 |
| Branch resolutions (N-FFT,Hop) | (512,256), (256,128), (128,64) | (640,320), (320,160), (160,80) |
| Branch embedding dimensions | 768,512,384 | 768,512,384 |
| Branch Layers | 8,8,8 | 8,8,8 |
| Feedforward hidden factor | 3 | 3 |
| Convolution kernel size | 7 | 7 |
| Condition encoder embedding dimension | 512 | 512 |
| Condition encoder layers | 4 | 4 |
| Loss scaling $\mathcal{S}(x)$, (N-FFT,Hop,N-filters) | (1024,256,256) | (1280,320,320) |

## A.4    RESULTS COMPARISON OF MEL AND LOG-MEL CONDITIONING

Table 11 compares Flow2GAN results with Mel and log-Mel conditioning on the LibriTTS test set.

Table 11: Comparison of Mel and log-Mel conditioning on LibriTTS test set.

| Model | Condition | PESQ↑ | ViSQOL↑ | V/UV F1↑ | Periodicity↓ | FSD↓ |
|---|---|---|---|---|---|---|
| Flow2GAN, 1-step | Mel | **4.288** | 4.939 | **0.976** | **0.058** | **0.027** |
| Flow2GAN, 1-step | log-Mel | 4.189 | **4.957** | 0.975 | 0.063 | 0.028 |
| Flow2GAN, 2-step | Mel | **4.454** | 4.966 | 0.981 | 0.045 | **0.019** |
| Flow2GAN, 2-step | log-Mel | 4.440 | **4.979** | **0.983** | **0.044** | 0.023 |
| Flow2GAN, 4-step | Mel | **4.484** | 4.966 | 0.984 | 0.039 | 0.017 |
| Flow2GAN, 4-step | log-Mel | **4.484** | **4.986** | **0.985** | **0.037** | **0.016** |

