# OpenReview forum: "Flow2GAN: Hybrid Flow Matching and GAN with Multi-Resolution Network for Few-step High-Fidelity Audio Generation"
_ICLR.cc/2026/Conference — ICLR 2026 Poster_

### Official Review · Reviewer_X1ZZ · 2025-10-19

**Soundness:** 4
**Presentation:** 3
**Contribution:** 2
**Rating:** 4
**Confidence:** 5

**Summary:**

This paper presents Flow2GAN, a framework that combines flow matching and GAN for efficient one- and two-step neural vocoder generation. It employs a robust flow matching training strategy based on end-point estimation and spectral energy–adaptive loss scaling to achieve high-fidelity waveform synthesis.

**Strengths:**

[CFM Training Optimization for Robust Waveform Generation]

This paper replaces vector field estimation with end-point estimation to achieve more robust waveform generation. In addition, the loss is scaled by spectral energy. Table 3 demonstrates the effectiveness of the proposed optimization method for high-fidelity waveform synthesis.

However, the analysis in Figure 2 is limited to the spectral domain. While end-point estimation combined with spectral energy–based scaling can improve training efficiency in the early stages and enhance spectral-domain performance, it would be beneficial to further evaluate the model using waveform-level metrics. I understand that standardized waveform-level metrics are lacking, but I recommend including MOS evaluations in Tables 1 and 2 to provide a more comprehensive assessment of perceptual quality.

**Weaknesses:**

While I like the concept of this paper, my primary concern lies in its lack of novelty.

[End-Point Estimation]

Previous works such as PeriodWave and RFWave have already demonstrated the effectiveness of conditional flow matching for high-fidelity waveform generation. The proposed end-point estimation seems to be a relatively simple modification. Moreover, several recent models, including sCM, have adopted similar end-point estimation schemes for few-step generation.

[Spectral Energy Scaling]

Many works already adopt an energy-based scaling. PriorGrad uses a data-dependent prior based on energy, and they scale the loss with this energy. RFWave also adjusts the loss according to the target's standard deviation.

Please add further discussion with previous scaling methods. Also, I have a concern about that it only reflect the spectral domain information by removing other information. Did you encounter any issues related to this limitation?

[GAN Fine-tuning]

Honestly, I can not find any difference between your approach and PeriodWave-Turbo which has already shown the efficiency of GAN post-training for CFM-based waveform generation. The paper seems to overclaim novelty in Section 4.2 regarding this aspect.

**Questions:**

[Q1. Multi-Resolution Network]

Are the inverted outputs from different iSTFT branches simply added together or averaged?

[Q2. Further Steps with GAN Fine-Tuning]

Have you experimented with training four-step or more generators after GAN fine-tuning? It would be interesting to see how performance scales with additional sampling steps.

[Q3. EnCodec Audio Token Experiments]

Could you include PeriodWave-Turbo in Table 2? PeriodWave-Turbo released checkpoints for the same EnCodec-based experiments on their GitHub, so a direct comparison would strengthen your results.

[Q4. TTS Experiments]

Please consider adding TTS results comparing various neural vocoders. Since some vocoders might overfit to ground-truth Mel-spectrograms, conducting two-stage TTS experiments (The generated Mel from TTS Models to waveform) would enhance the quality of the paper.

---

> ### Author Response · Authors · 2025-11-27
> **Thanks and response to concerns (part 1)**
>
> We sincerely thank the reviewer for the valuable and insightful comments. Below, we address each of the reviewer's concerns in detail.
>
> > I recommend including MOS evaluations in Tables 1 and 2 to provide a more comprehensive assessment of perceptual quality.
>
> Thank you for the suggestion. We have added the MOS metric to Tables 1 and 2 in the revised manuscript.
>
> > [End-Point Estimation] Previous works such as PeriodWave and RFWave have already demonstrated the effectiveness of conditional flow matching for high-fidelity waveform generation. The proposed end-point estimation seems to be a relatively simple modification. Moreover, several recent models, including sCM, have adopted similar end-point estimation schemes for few-step generation.
>
>
> 1) **Distinction from prior Flow Matching-based audio generation works:**
> We first clarify the distinction from prior works such as PeriodWave and RFWave, which use standard Flow Matching for audio generation. **Our work improves Flow Matching specifically for audio modeling considering audio's distinctive properties:** 1) reformulating the training objective from velocity to endpoint prediction, circumventing challenges in velocity estimation for empty regions; 2) incorporating spectral energy-based loss weighting to emphasize perceptually important low-energy regions.
>
> 2) **Distinction from Consistency Models:**
> Then we clarify that **our endpoint estimation reformulation differs fundamentally from Consistency Models [1][2] in both goal and training procedure:**
> - **Objective:** We reformulate Flow Matching from velocity to endpoint estimation to address audio-specific challenges, particularly velocity estimation difficulties in silent regions, thereby improving Flow Matching performance for audio modeling. Consistency Models [1][2], by contrast, represent a fundamentally different generative model class that aims to learn a direct one-step mapping from noise to data [3], rather than improving an existing flow-based framework.
> - **Training procedure:** Our approach uses standard Flow Matching training with a single model forward pass. Consistency Models [1][2] require two forward passes at different timesteps $x_{t}$ and $x_{t'}$ with consistency constraints enforced between them, necessitating either a pretrained model (Consistency Distillation) or an EMA model (Consistency Training).
>
> [1] Song, Y., Dhariwal, P., Chen, M., & Sutskever, I. (2023). Consistency models.
>
> [2] Lu, C., & Song, Y. (2024). Simplifying, stabilizing and scaling continuous-time consistency models. arXiv preprint arXiv:2410.11081.
>
> [3] Boffi, N. M., Albergo, M. S., & Vanden-Eijnden, E. (2024). Flow map matching. arXiv preprint arXiv:2406.07507, 2(3), 9.
>
> > [Spectral Energy Scaling] Many works already adopt an energy-based scaling. PriorGrad uses a data-dependent prior based on energy, and they scale the loss with this energy. RFWave also adjusts the loss according to the target's standard deviation. Please add further discussion with previous scaling methods.
>
> For our proposed spectral energy-adaptive loss scaling, the main difference from PriorGrad and RFWave is that our scaling considers both time and frequency dimensions, whereas PriorGrad and RFWave employ per-frame scaling that ignores energy differences across frequency regions. In the original manuscript, we mentioned this distinction in the Related Work section (Lines 106–107) and conducted an ablation study with results shown in Table 3. The results demonstrate that per-frame energy-based scaling, as used in PriorGrad and RFWave, results in performance degradation compared to our scaling method. In the revised manuscript, we have clarified this distinction more explicitly in Section 4.1 (Method) and Section 5.3 (Ablation study).
> | Scaling method | Flow Matching 2 steps  |GAN finetuned 1-step | GAN finetuned 2-step |
> | - | - | - | - |
> | per-frame loss scaling | PESQ=3.140,ViSQOL=3.667 | PESQ=4.201,ViSQOL=4.910 | PESQ=4.388,ViSQOL=4.942 |
> | Ours | PESQ=**3.469**,ViSQOL=**3.749** | PESQ=**4.303**,ViSQOL=**4.942** | PESQ=**4.471**,ViSQOL=**4.969** |
>
> > Also, I have a concern about that it only reflect the spectral domain information by removing other information. Did you encounter any issues related to this limitation?
>
> The loss scaling in Equation 6 converts the waveform domain error $g_{\theta}(x_t, t|\mathbf{c}) - x_1$ to spectral domain $\mathcal{S}(g_{\theta}(x_t, t|\mathbf{c}) - x_1)$. This transformation is differentiable, allowing gradients to flow back to the waveform domain during backpropagation. Importantly, this spectral energy-adaptive scaling does not discard waveform domain information; rather, it reweights the loss to emphasize regions where spectral energy is lower.

---

> ### Author Response · Authors · 2025-11-27
> **Thanks and response to concerns (part 2)**
>
> > [GAN Fine-tuning] Honestly, I can not find any difference between your approach and PeriodWave-Turbo which has already shown the efficiency of GAN post-training for CFM-based waveform generation. The paper seems to overclaim novelty in Section 4.2 regarding this aspect.
>
> Our method shares a similar high-level idea with PeriodWave-Turbo—using GAN fine-tuning to accelerate flow-based models. However, the key novelty lies in our **improved Flow Matching formulation specifically designed for audio**, which incorporates two critical enhancements: **endpoint estimation reformulation** and **spectral energy-adaptive loss scaling**. These improvements provide significantly stronger pretrained weights for GAN generator initialization compared to standard Flow Matching. As demonstrated in Table 3, using standard Flow Matching (as in PeriodWave-Turbo) results in substantially lower one-step result after GAN fine-tuning, whereas our improved Flow Matching achieves superior results. Additionally, PeriodWave-Turbo does not explore one-step generation, which is also a key focus of our work.
> | Pretrain | GAN finetuned 1-step | GAN finetuned 2-step |
> | - | - | - |
> | Standard Flow Matching | PESQ=3.730,ViSQOL=4.853 | PESQ=4.257,ViSQOL=4.933 |
> | Our improved Flow Matching | PESQ=**4.303**,ViSQOL=**4.942** | PESQ=**4.471**,ViSQOL=**4.969** |
>
> In the revised manuscript, we have clarified these distinctions more explicitly in Section 2 (Related Work), Section 4.2 (Method), and Section 5.3 (Ablation Studies), to better articulate how our improved Flow Matching objective, rather than the GAN fine-tuning strategy alone, drives our performance gains.
>
> > [Q1. Multi-Resolution Network] Are the inverted outputs from different iSTFT branches simply added together or averaged?
>
> As we described in the original manuscript, Line 288-L289 "The final output is obtained by summing all branch outputs."
>
> > [Q2. Further Steps with GAN Fine-Tuning] Have you experimented with training four-step or more generators after GAN fine-tuning? It would be interesting to see how performance scales with additional sampling steps.
>
> Thank you for this constructive suggestion. In the revised manuscript, we have added four-step generation results.
>
> For Mel-spectrogram conditioning (Table 1), the 4-step model outperforms both 1-step and 2-step variants on all metrics:
>
> | Model | PESQ$\uparrow$ | ViSQOL$\uparrow$ | V/UV F1$\uparrow$ | Periodicity$\downarrow$ | FSD$\downarrow$ | SMOS$\uparrow$ | MOS$\uparrow$ |
> |-|-|-|-|-|-|-|-|
> | Flow2GAN, 1-step | 4.189 | 4.957 | 0.975 | 0.063 | 0.028 | 4.44 $\pm$ 0.14 | 4.39 $\pm$ 0.15 |
> | Flow2GAN, 2-step | 4.440 | 4.979 | 0.983 | 0.044 | 0.023 | 4.53 $\pm$ 0.13 | 4.56 $\pm$ 0.11 |
> | Flow2GAN, 4-step | **4.484** | **4.986** | **0.985** | **0.037** | **0.016** | **4.60 $\pm$ 0.14** | **4.58 $\pm$ 0.14** |
>
> For Encodec audio token conditioning (Table 2), the 4-step model outperforms 1-step and 2-step variants on all objective metrics.
> | Bandwidth (kbps) | Model | PESQ$\uparrow$ |ViSQOL$\uparrow$ | FSD $\downarrow$ | SMOS$\uparrow$ | MOS$\uparrow$ |
> |-|-|-|-|-|-|-|
> | 1.5 | Flow2GAN, 1-step | 1.739 | 3.582 | 1.210 | 2.43 $\pm$ 0.20 | 2.83 $\pm$ 0.21 |
> | 1.5 | Flow2GAN, 2-step | 1.803 | 3.609 | 1.152 | 3.04 $\pm$ 0.20 | **3.86 $\pm$ 0.14** |
> | 1.5 | Flow2GAN, 4-step | **1.925** | **3.662** | **1.069**| **3.17 $\pm$ 0.19** | 3.40 $\pm$ 0.18 |
> |3.0| Flow2GAN, 1-step | 2.353 | 4.026 | 0.867 | 3.94 $\pm$ 0.14 | 4.00 $\pm$ 0.19 |
> |3.0| Flow2GAN, 2-step | 2.442 | 4.049 | 0.843 | **4.19 $\pm$ 0.15** | 4.07 $\pm$ 0.24 |
> |3.0| Flow2GAN, 4-step | **2.550** | **4.091** | **0.804** | 4.03 $\pm$ 0.16 | **4.08 $\pm$ 0.22** |
> |6.0| Flow2GAN, 1-step | 2.904 | 4.300 | 0.696 | **4.46 $\pm$ 0.16** | **4.42 $\pm$ 0.22**|
> |6.0| Flow2GAN, 2-step | 2.983 | 4.319 | 0.733 | 4.38 $\pm$ 0.14 | 4.31 $\pm$ 0.17 |
> |6.0| Flow2GAN, 4-step | **3.089** | **4.351** | **0.678** | 4.19 $\pm$ 0.12 | 4.38 $\pm$ 0.13|
> |12.0| Flow2GAN, 1-step | 3.389 | 4.489 | 0.632 | **4.52 $\pm$ 0.13** | **4.53 $\pm$ 0.23** |
> |12.0| Flow2GAN, 2-step | 3.457 | 4.505 | 0.600 | 4.39 $\pm$ 0.14 | 4.44 $\pm$ 0.28 |
> |12.0| Flow2GAN, 4-step | **3.538** | **4.531** | **0.557** | 4.22 $\pm$ 0.19 | 4.50 $\pm$ 0.15 |

---

> ### Author Response · Authors · 2025-11-27
> **Thanks and response to concerns (part 3)**
>
> > [Q3. EnCodec Audio Token Experiments] Could you include PeriodWave-Turbo in Table 2? PeriodWave-Turbo released checkpoints for the same EnCodec-based experiments on their GitHub, so a direct comparison would strengthen your results.
>
> Thanks for pointing out this. In the revised manuscript, we have added these Encodec-based results in Table 2, using the release checkpoint on the its Github.
>
> At lower bandwidths of 1.5 and 3.0 kbps, all Flow2GAN variants outperform it across most metrics, except that the one-step and two-step models underperform PeriodWave-Turbo in ViSQOL at 3.0 kbps. At higher bandwidths of 6.0 and 12.0 kbps, although PeriodWave-Turbo achieves the best results in PESQ and ViSQOL, the four-step Flow2GAN achieving notably better FSD results than PeriodWave-Turbo.
>
> | Bandwidth (kbps) | Model | PESQ$\uparrow$ |ViSQOL$\uparrow$ | FSD $\downarrow$ | SMOS$\uparrow$ | MOS$\uparrow$ |
> |-|-|-|-|-|-|-|
> | 1.5 | PeriodWave-Turbo (4-step) | 1.260 | 3.308 | 4.055 | 1.55 $\pm$ 0.16 | 1.47 $\pm$ 0.19|
> | 1.5 | Flow2GAN, 1-step | 1.739 | 3.582 | 1.210 | 2.43 $\pm$ 0.20 | 2.83 $\pm$ 0.21 |
> | 1.5 | Flow2GAN, 2-step | 1.803 | 3.609 | 1.152 | 3.04 $\pm$ 0.20 | **3.86 $\pm$ 0.14** |
> | 1.5 | Flow2GAN, 4-step | **1.925** | **3.662** | **1.069**| **3.17 $\pm$ 0.19** | 3.40 $\pm$ 0.18 |
> |3.0 | PeriodWave-Turbo (4-step) | 2.160 | 4.058 | 1.018 | 3.04 $\pm$ 0.17 | 3.16 $\pm$ 0.23 |
> |3.0| Flow2GAN, 1-step | 2.353 | 4.026 | 0.867 | 3.94 $\pm$ 0.14 | 4.00 $\pm$ 0.19 |
> |3.0| Flow2GAN, 2-step | 2.442 | 4.049 | 0.843 | **4.19 $\pm$ 0.15** | 4.07 $\pm$ 0.24 |
> |3.0| Flow2GAN, 4-step | **2.550** | **4.091** | **0.804** | 4.03 $\pm$ 0.16 | **4.08 $\pm$ 0.22** |
> |6.0| PeriodWave-Turbo (4-step) | **3.229** | **4.424** | 0.712 | 4.00 $\pm$ 0.17 | 4.40 $\pm$ 0.21 |
> |6.0| Flow2GAN, 1-step | 2.904 | 4.300 | 0.696 | **4.46 $\pm$ 0.16** | **4.42 $\pm$ 0.22**|
> |6.0| Flow2GAN, 2-step | 2.983 | 4.319 | 0.733 | 4.38 $\pm$ 0.14 | 4.31 $\pm$ 0.17 |
> |6.0| Flow2GAN, 4-step | 3.089 | 4.351 | **0.678** | 4.19 $\pm$ 0.12 | 4.38 $\pm$ 0.13|
> |12.0| PeriodWave-Turbo (4-step) | **3.579** | **4.544** | 0.776 | 4.32 $\pm$ 0.16 | **4.56 $\pm$ 0.16**|
> |12.0| Flow2GAN, 1-step | 3.389 | 4.489 | 0.632 | **4.52 $\pm$ 0.13** | 4.53 $\pm$ 0.23 |
> |12.0| Flow2GAN, 2-step | 3.457 | 4.505 | 0.600 | 4.39 $\pm$ 0.14 | 4.44 $\pm$ 0.28 |
> |12.0| Flow2GAN, 4-step | 3.538 | 4.531 | **0.557** | 4.22 $\pm$ 0.19 | 4.50 $\pm$ 0.15 |

---

> ### Author Response · Authors · 2025-11-27
> **Thanks and response to concerns (part 4)**
>
> > [Q4. TTS Experiments] Please consider adding TTS results comparing various neural vocoders. Since some vocoders might overfit to ground-truth Mel-spectrograms, conducting two-stage TTS experiments (The generated Mel from TTS Models to waveform) would enhance the quality of the paper.
>
> Thank you for the constructive suggestion. We have added zero-shot TTS results comparing various vocoders in Section 5.5. Specifically, we use the recent zero-shot TTS model, F5-TTS Base (https://github.com/SWivid/F5-TTS). Note that due to differences in mel-spectrogram implementations between librosa and torchaudio, it releases two separate TTS models. We test the zero-shot TTS performance on LibriSpeech-PC test-clean set.
>
> **Change conditioning from Mel to log-Mel**: Since F5-TTS generates log-Mel spectrograms, we have switched our conditioning from Mel to log-Mel to better match it and for fair comparison. We present Flow2GAN results with both Mel and log-Mel conditioning on the LibriTTS test set in Appendix Table 11. Since log-Mel yields slight improvements, we have updated Table 1 (comparisons with other models) to use log-Mel conditioning, while maintaining Mel conditioning for ablation studies.
>
> |Model | Condition | PESQ$\uparrow$ | ViSQOL$\uparrow$ | V/UV F1$\uparrow$ | Periodicity$\downarrow$ | FSD$\downarrow$ |
> |-|-|-|-|-|-|-|
> |Flow2GAN, 1-step | Mel | **4.288** | 4.939 | **0.976** | **0.058** | **0.027** |
> | Flow2GAN, 1-step | log-Mel | 4.189 | **4.957** | 0.975 | 0.063 | 0.028 |
> | Flow2GAN, 2-step | Mel | **4.454** | 4.966 | 0.981 | 0.045 | **0.019** |
> |  Flow2GAN, 2-step | log-Mel | 4.440 | **4.979** | **0.983** | **0.044** | 0.023 |
> | Flow2GAN, 4-step  | Mel | **4.484** | 4.966 | 0.984 | 0.039 | 0.017 |
> | Flow2GAN, 4-step  | log-Mel | **4.484** | **4.986** | **0.985** | **0.037** | **0.016** |
>
> **TTS results**: Our 4-step Flow2GAN achieves comparable SIM-o and superior UTMOS relative to PeriodWave-Turbo, with a substantial speed advantage (Table 7), despite higher WER. An interesting observation: BigVGAN-v2 underperforms BigVGAN, indicating its possible overfitting to ground-truth Mel-spectrograms. Interestingly, we found that adding small Gaussian noise to the conditioning log-Mel spectrograms in GAN fine-tuning, improves WER and UTMOS,  possibly since it increases model robustness to imperfect TTS outputs.
>
> |Model |  Mel-type | WER(%)$\downarrow$ | SIM-o$\uparrow$ | UTMOS$\uparrow$ |
> |-|-|-|-|-|
> | BigVGAN |  librosa | **1.78** | 0.67 | 4.05 |
> |BigVGAN-v2 | librosa | 1.82 | 0.66 | 3.70 |
> | PeriodWave-Turbo (4 steps) | librosa | 1.8 | **0.67** | 4.17 |
> | Vocos | torchaudio | 1.91 | 0.65 | 3.90 |
> | RFWave (10 steps) | torchaudio|  1.87 | 0.66 | 3.63 |
> | WaveFM (1 step) | torchaudio| 2.01 | 0.65 | 3.15 |
> | Flow2GAN, 1-step | torchaudio | 1.94 | **0.67** | 3.75 |
> | Flow2GAN, 2-step | torchaudio | 1.91 | **0.67** | 4.11 |
> | Flow2GAN, 4-step | torchaudio | 1.89 | **0.67** | 4.18 |
> | Flow2GAN, 1-step$^{\dagger}$ | torchaudio | 1.88 | **0.67** | 3.83 |
> | Flow2GAN, 2-step$^{\dagger}$ | torchaudio | 1.88 | **0.67** | 4.21 |
> | Flow2GAN, 4-step$^{\dagger}$ | torchaudio | 1.87 | **0.67** | **4.26** |
> Note: $^{\dagger}$ denotes adding noise to log-Mel during GAN fine-tuning.

---

### Official Review · Reviewer_Vxds · 2025-10-31

**Soundness:** 2
**Presentation:** 3
**Contribution:** 2
**Rating:** 6
**Confidence:** 3

**Summary:**

Flow2GAN introduces a two-stage framework combining Flow Matching (FM) pre-training and GAN fine-tuning to balance audio generation quality and inference speed. Key innovations include endpoint-prediction-based FM (to handle silent regions), spectral energy-adaptive loss scaling (for perceptual alignment), and a multi-resolution ConvNeXt network. Experiments show competitive performance against SOTA models, though critical gaps exist in comparisons to key competitors and hybrid FM+GAN baselines.

**Strengths:**

Hybrid Paradigm: Effectively merges FM’s stable training with GAN’s detail refinement, resolving FM’s slow inference and GAN’s mode collapse issues.
Audio-Specific FM Improvements: Endpoint prediction and energy-adaptive loss significantly boost FM’s performance, validating their utility for audio synthesis.
Strong Experimental Results: Outperforms Vocos, RFWave, and WaveFM on most metrics (Table1/2) for Mel-spectrogram and Encodec token conditioning.
Multi-Resolution Network: Enhances frequency modeling compared to single-resolution designs (Table5), improving perceptual audio quality.

**Weaknesses:**

Training Complexity: Two-stage training (FM pre-training + GAN fine-tuning) increases the barrier to reproduction and deployment—users need to manage separate pipelines and hyperparameters for each stage.

Incomplete and Unconvincing Comparison Landscape:
a. Lack of coverage of GAN-enhanced Flow Matching models: The paper claims novelty in its hybrid FM+GAN framework, but many existing models leverage GANs to accelerate Flow Matching. A detailed comparison to these models—including their design choices, performance, and efficiency—is missing, which obscures Flow2GAN’s unique contributions and relative standing.
b. Performance does not exceed BigVGAN: Even accounting for dataset size differences (BigVGAN uses a larger dataset), the paper’s reported metrics (e.g., PESQ, audio quality scores) do not demonstrate that Flow2GAN outperforms BigVGAN. The claim of being "state-of-the-art" is thus unsubstantiated against this key competitor.
c. Significant speed gap with Vocos: The paper emphasizes inference efficiency but fails to address that its speed is much slower than Vocos (a leading efficient audio generation model). This gap undermines the practical utility of Flow2GAN for real-world applications where low latency is critical.

**Questions:**

NA

---

> ### Author Response · Authors · 2025-11-27
> **Thanks and response to concerns (part 1)**
>
> We sincerely thank the reviewer for the valuable and insightful comments. Below, we address each of the reviewer's concerns in detail.
>
> > Training Complexity: Two-stage training (FM pre-training + GAN fine-tuning) increases the barrier to reproduction and deployment—users need to manage separate pipelines and hyperparameters for each stage.
>
> We appreciate this concern. However, our two-stage approach (improved Flow Matching pre-training + lightweight GAN fine-tuning) actually reduces overall training cost and achieves better results compared to one-stage pure GAN training, as demonstrated in Table 4. To further facilitate reproducibility, we will release our code and checkpoints upon acceptance, enabling users to easily reproduce and build upon our work. We have also added a Reproducibility Statement in the revised manuscript.

---

> ### Author Response · Authors · 2025-11-27
> **Thanks and response to concerns (part 2)**
>
> >  a. Lack of coverage of GAN-enhanced Flow Matching models: The paper claims novelty in its hybrid FM+GAN framework, but many existing models leverage GANs to accelerate Flow Matching. A detailed comparison to these models—including their design choices, performance, and efficiency—is missing, which obscures Flow2GAN’s unique contributions and relative standing.
>
> For existing model leveraging GANs to accelerate Flow Matching in audio generation, the recent work PeriodWave-Turbo is most relevant. In the original manuscript, we compared against it in Mel-conditioned audio generation quality, and inference speed. Following Reviewer X1ZZ's suggestions, in the revised version, we have added: 4-step Flow2GAN results, TTS vocoder comparisons, and PeriodWave-Turbo's Encodec-based results. Since the TTS model generates log-Mel spectrograms, we have switched our conditioning from Mel to log-Mel to better match it and for fair comparison.
>
> Now we summary our comparisons to PeriodWave-Turbo:
>
> 1) **Mel-conditioned audio generation (Table 1)**: our 2-step models outperforms it in most metrics except FSD and our 4-step model outperforms it in all metrics.
>
> | Model | PESQ$\uparrow$ | ViSQOL$\uparrow$ | V/UV F1$\uparrow$ | Periodicity$\downarrow$ | FSD$\downarrow$ | SMOS$\uparrow$ | MOS$\uparrow$ |
> |-|-|-|-|-|-|-|-|
> | PeriodWave-Turbo | 4.434 | 4.965 | 0.958 | 0.096 | 0.020 | 4.20 $\pm$ 0.17 | 4.38 $\pm$  0.17 |
> | Flow2GAN, 1-step | 4.189 | 4.957 | 0.975 | 0.063 | 0.028 | 4.44 $\pm$ 0.14 | 4.39 $\pm$ 0.15 |
> | Flow2GAN, 2-step | 4.440 | 4.979 | 0.983 | 0.044 | 0.023 | 4.53 $\pm$ 0.13 | 4.56 $\pm$ 0.11 |
> | Flow2GAN, 4-step | **4.484** | **4.986** | **0.985** | **0.037** | **0.016** | **4.60 $\pm$ 0.14** | **4.58 $\pm$ 0.14** |
>
> 2) **Encodec token-conditioned audio generation (Table 2)**: At lower bandwidths of 1.5 and 3.0 kbps, all Flow2GAN variants outperform PeriodWave-Turbo across most metrics, except that the 1-step and 2-step models underperform PeriodWave-Turbo in ViSQOL at 3.0 kbps. At higher bandwidths of 6.0 and 12.0 kbps, though PeriodWave-Turbo achieves the best PESQ and ViSQOL, the four-step Flow2GAN achieves better FSD than PeriodWave-Turbo.
>
> | Bandwidth (kbps) | Model | PESQ$\uparrow$ |ViSQOL$\uparrow$ | FSD $\downarrow$ | SMOS$\uparrow$ | MOS$\uparrow$ |
> |-|-|-|-|-|-|-|
> | 1.5 | PeriodWave-Turbo (4-step) | 1.260 | 3.308 | 4.055 | 1.55 $\pm$ 0.16 | 1.47 $\pm$ 0.19|
> | 1.5 | Flow2GAN, 1-step | 1.739 | 3.582 | 1.210 | 2.43 $\pm$ 0.20 | 2.83 $\pm$ 0.21 |
> | 1.5 | Flow2GAN, 2-step | 1.803 | 3.609 | 1.152 | 3.04 $\pm$ 0.20 | **3.86 $\pm$ 0.14** |
> | 1.5 | Flow2GAN, 4-step | **1.925** | **3.662** | **1.069**| **3.17 $\pm$ 0.19** | 3.40 $\pm$ 0.18 |
> |3.0 | PeriodWave-Turbo (4-step) | 2.160 | 4.058 | 1.018 | 3.04 $\pm$ 0.17 | 3.16 $\pm$ 0.23 |
> |3.0| Flow2GAN, 1-step | 2.353 | 4.026 | 0.867 | 3.94 $\pm$ 0.14 | 4.00 $\pm$ 0.19 |
> |3.0| Flow2GAN, 2-step | 2.442 | 4.049 | 0.843 | **4.19 $\pm$ 0.15** | 4.07 $\pm$ 0.24 |
> |3.0| Flow2GAN, 4-step | **2.550** | **4.091** | **0.804** | 4.03 $\pm$ 0.16 | **4.08 $\pm$ 0.22** |
> |6.0| PeriodWave-Turbo (4-step) | **3.229** | **4.424** | 0.712 | 4.00 $\pm$ 0.17 | 4.40 $\pm$ 0.21 |
> |6.0| Flow2GAN, 1-step | 2.904 | 4.300 | 0.696 | **4.46 $\pm$ 0.16** | **4.42 $\pm$ 0.22**|
> |6.0| Flow2GAN, 2-step | 2.983 | 4.319 | 0.733 | 4.38 $\pm$ 0.14 | 4.31 $\pm$ 0.17 |
> |6.0| Flow2GAN, 4-step | 3.089 | 4.351 | **0.678** | 4.19 $\pm$ 0.12 | 4.38 $\pm$ 0.13|
> |12.0| PeriodWave-Turbo (4-step) | **3.579** | **4.544** | 0.776 | 4.32 $\pm$ 0.16 | **4.56 $\pm$ 0.16**|
> |12.0| Flow2GAN, 1-step | 3.389 | 4.489 | 0.632 | **4.52 $\pm$ 0.13** | 4.53 $\pm$ 0.23 |
> |12.0| Flow2GAN, 2-step | 3.457 | 4.505 | 0.600 | 4.39 $\pm$ 0.14 | 4.44 $\pm$ 0.28 |
> |12.0| Flow2GAN, 4-step | 3.538 | 4.531 | **0.557** | 4.22 $\pm$ 0.19 | 4.50 $\pm$ 0.15 |
>
> 3) **Speed comparison (Table 7)**: our models are much faster than PeriodWave-Turbo on both CPU and GPU.
> | Model | CPU xRT$\uparrow$ | GPU xRT$\uparrow$ |
> | - | - | - |
> | PeriodWave-Turbo (4-step) | 0.12 | 43.70 |
> | Flow2GAN, 1-step | 4.85 | 851.67 |
> | Flow2GAN, 2-step | 2.46 | 449.26 |
> | Flow2GAN, 4-step | 1.35 | 228.48 |
>
> **(continued below)**

---

> > ### Author Response · Authors · 2025-11-27
> > **Thanks and response to concerns (part 2) - Continued**
> >
> > 4) Key difference from PeriodWave-Turbo:
> > While our method shares the GAN fine-tuning strategy with PeriodWave-Turbo, the key distinction is our **improved Flow Matching formulation designed specifically for audio modeling**. Our enhancements—**endpoint estimation reformulation** and **spectral energy-adaptive loss scaling**—provide substantially stronger pretrained weights for GAN initialization. Ablation Table 3 demonstrates this advantage: standard Flow Matching (as in PeriodWave-Turbo) yields lower one-step performance after GAN fine-tuning, whereas our approach achieves superior results. Additionally, PeriodWave-Turbo does not explore one-step generation, which is a focus of our work.
> >
> > | Pretrain | GAN finetuned 1-step | GAN finetuned 2-step |
> > | - | - | - |
> > | Standard Flow Matching | PESQ=3.730,ViSQOL=4.853 | PESQ=4.257,ViSQOL=4.933 |
> > | Our improved Flow Matching | PESQ=**4.303**,ViSQOL=**4.942** | PESQ=**4.471**,ViSQOL=**4.969** |
> >
> > We hope above comparison on overall results, inference speed, and key method difference address your concerns.

---

> ### Author Response · Authors · 2025-11-27
> **Thanks and response to concerns (part 3)**
>
> > b. Performance does not exceed BigVGAN: Even accounting for dataset size differences (BigVGAN uses a larger dataset), the paper’s reported metrics (e.g., PESQ, audio quality scores) do not demonstrate that Flow2GAN outperforms BigVGAN. The claim of being "state-of-the-art" is thus unsubstantiated against this key competitor.
>
>
> Following Reviewer X1ZZ's suggestions, we have expanded the experimental evaluation with TTS vocoder comparisons and 4-step Flow2GAN results. To better match the TTS model which generates log-Mel spectrograms, we switched conditioning from Mel to log-Mel, resulting in slight performance improvements reflected in the updated Table 1. Detailed Mel vs. log-Mel ablation results are provided in Appendix Table 11.
>
> |Model | Condition | PESQ$\uparrow$ | ViSQOL$\uparrow$ | V/UV F1$\uparrow$ | Periodicity$\downarrow$ | FSD$\downarrow$ |
> |-|-|-|-|-|-|-|
> |Flow2GAN, 1-step | Mel | **4.288** | 4.939 | **0.976** | **0.058** | **0.027** |
> | Flow2GAN, 1-step | log-Mel | 4.189 | **4.957** | 0.975 | 0.063 | 0.028 |
> | Flow2GAN, 2-step | Mel | **4.454** | 4.966 | 0.981 | 0.045 | **0.019** |
> |  Flow2GAN, 2-step | log-Mel | 4.440 | **4.979** | **0.983** | **0.044** | 0.023 |
> | Flow2GAN, 4-step  | Mel | **4.484** | 4.966 | 0.984 | 0.039 | 0.017 |
> | Flow2GAN, 4-step  | log-Mel | **4.484** | **4.986** | **0.985** | **0.037** | **0.016** |
>
> Our 2-step and 4-step Flow2GAN models with log-Mel conditioning now outperform BigVGAN-v2 across most metrics, including PESQ, ViSQOL, V/UV F1, and Periodicity.
> | Model | PESQ$\uparrow$ | ViSQOL$\uparrow$ | V/UV F1$\uparrow$ | Periodicity$\downarrow$ | FSD$\downarrow$ | SMOS$\uparrow$ | MOS$\uparrow$ |
> |-|-|-|-|-|-|-|-|
> | BigVGAN-v2 | 4.379 | 4.971 | 0.978 | 0.055 | **0.014** | **4.65 $\pm$ 0.11** | **4.59 $\pm$ 0.10** |
> | Flow2GAN, 1-step | 4.189 | 4.957 | 0.975 | 0.063 | 0.028 | 4.44 $\pm$ 0.14 | 4.39 $\pm$ 0.15 |
> | Flow2GAN, 2-step | 4.440 | 4.979 | 0.983 | 0.044 | 0.023 | 4.53 $\pm$ 0.13 | 4.56 $\pm$ 0.11 |
> | Flow2GAN, 4-step | **4.484** | **4.986** | **0.985** | **0.037** | 0.016 | 4.60 $\pm$ 0.14 | 4.58 $\pm$ 0.14 |
>
> For TTS performance, all Flow2GAN models outperform BigVGAN-v2 in both SIM-o and UTMOS, with the 2-step and 4-step variants achieving notably better UTMOS:
>
> |Model |  Mel-type | WER(%)$\downarrow$ | SIM-o$\uparrow$ | UTMOS$\uparrow$ |
> |-|-|-|-|-|
> |BigVGAN-v2 | librosa | **1.82** | 0.66 | 3.70 |
> | Flow2GAN, 1-step | torchaudio | 1.94 | **0.67** | 3.75 |
> | Flow2GAN, 2-step | torchaudio | 1.91 | **0.67** | 4.11 |
> | Flow2GAN, 4-step | torchaudio | 1.89 | **0.67** | 4.18 |
> | Flow2GAN, 1-step$^{\dagger}$ | torchaudio | 1.88 | **0.67** | 3.83 |
> | Flow2GAN, 2-step$^{\dagger}$ | torchaudio | 1.88 | **0.67** | 4.21 |
> | Flow2GAN, 4-step$^{\dagger}$ | torchaudio | 1.87 | **0.67** | **4.26** |
> Note: $^{\dagger}$ denotes adding noise to log-Mel during GAN fine-tuning.
>
> > c. Significant speed gap with Vocos: The paper emphasizes inference efficiency but fails to address that its speed is much slower than Vocos (a leading efficient audio generation model). This gap undermines the practical utility of Flow2GAN for real-world applications where low latency is critical.
>
> We appreciate this concern. However, except for Vocos, whose generation quality is lower (Table 1), all Flow2GAN variants achieve significantly faster inference than higher-quality competing models—including BigVGAN-v2, RFWave, and PeriodWave-Turbo—on both CPU and GPU, **even achieving real-time performance on CPU**. This demonstrates that Flow2GAN achieves superior quality-efficiency trade-offs compared to existing methods.
>
> | Model | CPU xRT$\uparrow$ | GPU xRT$\uparrow$ |
> | - | - | - |
> | BigVGAN-v2 | 0.214 | 69.8 |
> | RFWave (10-step)  | 0.37 | 158.8 |
> | PeriodWave-Turbo (4-step) | 0.12 | 43.70 |
> | Flow2GAN, 1-step | 4.85 | 851.67 |
> | Flow2GAN, 2-step | 2.46 | 449.26 |
> | Flow2GAN, 4-step | 1.35 | 228.48 |
>
> In the revised manuscript, we have improved our statements and emphasized the quality-efficiency trade-offs of our model throughout the abstract, introduction, and conclusion to present the manuscript more rigorously.

---

### Official Review · Reviewer_AhmV · 2025-10-31

**Soundness:** 3
**Presentation:** 3
**Contribution:** 3
**Rating:** 6
**Confidence:** 4

**Summary:**

This paper introduces Flow2GAN, a two-stage hybrid framework for high-fidelity neural vocoding that aims to combine the training stability of Flow Matching with the detail-refinement capabilities of GANs. The authors first propose improvements to the Flow Matching stage tailored for audio, including reformulating the objective to direct endpoint prediction and introducing a spectral energy-adaptive loss scaling to focus on perceptually important regions. Subsequently, a pre-trained model is used to initialize one- and two-step generators, which are then fine-tuned using adversarial training. The architecture is a multi-resolution network that processes spectral coefficients at different time-frequency scales. Experiments on both mel-spectrogram and audio token conditioning show that Flow2GAN achieves state-of-the-art results in terms of both audio quality and inference efficiency.

**Strengths:**

The paper's core concept of a two-stage training paradigm is well-motivated and presents a clever solution to a known trade-off in generative modeling. The approach logically leverages Flow Matching for robust, global structure learning and then uses a fast GAN fine-tuning stage for refining high-frequency details, which is an effective strategy. The proposed modifications to the Flow Matching objective appear sound; the shift to endpoint prediction is an intuitive way to handle silent regions in audio, and the ablation studies convincingly demonstrate its benefits. The experimental results are another clear strength. The model achieves impressive scores across multiple metrics and tasks, outperforming several strong baselines while offering significantly faster inference than multi-step diffusion models.

**Weaknesses:**

Despite the strong results, the paper has several weaknesses in its positioning and methodological clarity that should be addressed. First, the proposed "spectral energy-adaptive loss scaling" is conceptually very similar to the "energy balanced loss" used in prior work like RFWave, yet the paper fails to discuss, or compare against it. This omission makes it difficult to assess the novelty of this specific contribution. Second, the reformulation of the prediction target from velocity to endpoint is a significant change to the underlying probability flow ODE. The paper presents a modified sampling equation but does not provide a clear derivation or justification for it, leaving a gap in the methodological explanation.
Furthermore, a key concern with any GAN-based method is the risk of mode collapse. The paper claims that its two-stage approach mitigates this risk, but this assertion is made without any supporting evidence, either quantitative (e.g., diversity metrics) or qualitative. Finally, while the paper emphasizes its fast one/two-step inference, it critically lacks comparisons to other state-of-the-art fast sampling methods for flow and diffusion models, such as consistency models or recent shortcut models, which are the most relevant competitors for few-step generative performance.

**Questions:**

To help clarify the contributions and rigor of the paper, I would appreciate the authors' response to the following:
1. Could you please elaborate on the relationship between your proposed spectral energy-adaptive loss scaling and the energy balanced loss from RFWave? How does your approach differ, and what are its specific advantages that lead to the observed performance gains?
2. When you reformulate the training objective to endpoint prediction (Equation 4), the underlying ODE changes. Could you provide a more thorough derivation or explanation for the new sampling process?
3. You claim that the Flow Matching pre-training mitigates the risk of GAN mode collapse. This is a significant claim. Could you provide any empirical evidence to support it, for instance, by analyzing the output diversity of Flow2GAN compared to a pure GAN baseline trained for a similar duration?
4. The main appeal of the model is high-quality generation in one or two steps. Why were there no comparisons against other prominent few-step or single-step generative model sampling strategies, such as consistency models or recent "shortcut models" (e.g., as proposed by Frans et al., 2024), which are designed to solve the exact same problem?

---

> ### Author Response · Authors · 2025-11-27
> **Thanks and response to concerns (part 1)**
>
> We sincerely thank the reviewer for the valuable and insightful comments. Below, we address each of the reviewer's concerns in detail.
>
> > First, the proposed "spectral energy-adaptive loss scaling" is conceptually very similar to the "energy balanced loss" used in prior work like RFWave, yet the paper fails to discuss, or compare against it. This omission makes it difficult to assess the novelty of this specific contribution.
>
> > Could you please elaborate on the relationship between your proposed spectral energy-adaptive loss scaling and the energy balanced loss from RFWave? How does your approach differ, and what are its specific advantages that lead to the observed performance gains?
>
> For our proposed spectral energy-adaptive loss scaling, the main difference from RFWave's is that our scaling is more comprehensive as **it considers both time and frequency dimensions**, whereas RFWave's scaling is **per-frame**, ignoring energy differences across frequency bands.
> In the original manuscript, we mentioned this difference in Related Work (Lines 106–107) and conducted an ablation study with results shown in Table 3. The results demonstrate that per-frame energy-based scaling, as in RFWave, leads to notable performance degradation compared to our method.
>
> | Scaling method | Flow Matching 2 steps  |GAN finetuned 1-step | GAN finetuned 2-step |
> | - | - | - | - |
> | Per-frame loss scaling | PESQ=3.140,ViSQOL=3.667 | PESQ=4.201,ViSQOL=4.910 | PESQ=4.388,ViSQOL=4.942 |
> | Ours | PESQ=**3.469**,ViSQOL=**3.749** | PESQ=**4.303**,ViSQOL=**4.942** | PESQ=**4.471**,ViSQOL=**4.969** |
>
> To address this confusion, in the revised manuscript, we have clarified the distinction from RFWave's per-frame scaling in Section 4.1 (Method) and analyzed its benefits in Section 5.3 (Ablation Analysis).
>
> > Second, the reformulation of the prediction target from velocity to endpoint is a significant change to the underlying probability flow ODE. The paper presents a modified sampling equation but does not provide a clear derivation or justification for it, leaving a gap in the methodological explanation.
>
> > When you reformulate the training objective to endpoint prediction (Equation 4), the underlying ODE changes. Could you provide a more thorough derivation or explanation for the new sampling process?
>
> We would like to clarify that, in the original manuscript, **the reformulation from velocity $v_t$ estimation (Equation 1) to endpoint $x_1$ estimation (Equation 3) does not change the underlying probability flow ODE**.
> For the transformation from Equation 1 to Equation 3,
> we just need to replace $f_{\theta}(x_t, t)$ with
> $\frac{g_{\theta}(x_t, t|\mathbf{c}) - x_t}{1 - t}$ and $v_t$ with
> $v_t = \frac{x_1 - x_t}{1 - t}$.
> **Equation 1 and Equation 3 are mathematically equivalent and describe the same probability flow, just expressed through different network prediction targets (velocity vs. endpoint).**
>
> The corresponding sampling equations are likewise equivalent. Equation 2 $x_{t_{i+1}} = x_{t_i} + (t_{i+1} - t_i)f_{\theta}(x_{t_i}, t_i)$ becomes to Equation 5 $x_{t_{i+1}} = x_{t_i} + (t_{i+1} - t_i)\frac{g_{\theta}(x_{t_i}, t_i|\mathbf{c}) - x_{t_i}}{1 - t_i}.$.
>
> In addition, as we mentioned in the original manuscript, we found that removing the weighting factor $\frac{1}{(1 - t)^2}$ in Equation 3 results in slight improvements. The reason might be that without this weighting, the model can better emphasize small $t$ values, which aligns well with our setting. Therefor, for simplicity, we omit this factor, finally getting Equation 4.
>
> If there are any further concerns about this, we welcome further discussion.
>
> > Furthermore, a key concern with any GAN-based method is the risk of mode collapse. The paper claims that its two-stage approach mitigates this risk, but this assertion is made without any supporting evidence, either quantitative (e.g., diversity metrics) or qualitative.
>
> > You claim that the Flow Matching pre-training mitigates the risk of GAN mode collapse. This is a significant claim. Could you provide any empirical evidence to support it, for instance, by analyzing the output diversity of Flow2GAN compared to a pure GAN baseline trained for a similar duration?
>
> Thank you for pointing out this assertion. We have removed the claim that our two-stage approach "mitigates the risk of mode collapse" for GAN training from Section 4.2 in the revised version, as it lacked rigorous supporting evidence and it is challenging to quantitatively measure such metrics (e.g., by diversity metrics) for GAN models.
> We would like to re-emphasize that our two-stage approach—combining improved Flow Matching pretraining with GAN fine-tuning—significantly reduces overall training cost compared to pure GAN training, as demonstrated in Section 4.2 and Section 5.3 Table 4.
> We hope this modification makes the paper more rigorous and addresses your concern.

---

> > ### Author Response · Authors · 2025-11-27
> > **Thanks and response to concerns (part 2)**
> >
> > > Finally, while the paper emphasizes its fast one/two-step inference, it critically lacks comparisons to other state-of-the-art fast sampling methods for flow and diffusion models, such as consistency models or recent shortcut models, which are the most relevant competitors for few-step generative performance.
> >
> > > The main appeal of the model is high-quality generation in one or two steps. Why were there no comparisons against other prominent few-step or single-step generative model sampling strategies, such as consistency models or recent "shortcut models" (e.g., as proposed by Frans et al., 2024), which are designed to solve the exact same problem?
> >
> > Thank you for this suggestion. In the revised manuscript, we have added experiments (Table 5) comparing our approach to ShortCut models, an advanced diffusion/Flow-Matching variant for few-step generation. Our results demonstrate that our Flow2GAN models outperform ShortCut models in audio generation across all few step counts (1, 2, and 4 steps).
> > - ShortCut models:
> >   - 1-step: PESQ=3.181, ViSQOL=4.426
> >   - 2-step: PESQ=3.841, ViSQOL=4.660
> >   - 4-step: PESQ=4.122, ViSQOL=4.781
> > - Flow2GAN:
> >   - 1-step: PESQ=**4.303**, ViSQOL=**4.942**
> >   - 2-step: PESQ=**4.471**, ViSQOL=**4.969**
> >   - 4-step: PESQ=**4.498**, ViSQOL=**4.967**

---

### Official Review · Reviewer_KvbH · 2025-11-01

**Soundness:** 3
**Presentation:** 3
**Contribution:** 3
**Rating:** 6
**Confidence:** 2

**Summary:**

Flow2GAN is a two-stage framework for high-fidelity audio generation that integrates Flow Matching and GAN. It first uses improved Flow Matching to learn robust generative capabilities, which reformulated as endpoint estimation to avoid velocity prediction issues in empty audio regions and enhanced with spectral energy-adaptive loss scaling to emphasize perceptually important quiet areas. Then, lightweight GAN fine-tuning refines details, enabling efficient one- or two-step inference. Equipped with a multi-resolution network processing Fourier coefficients at different time-frequency resolutions, it outperforms state-of-the-art GAN and Flow Matching-based methods in both quality and efficiency under Mel-spectrogram and Encodec audio token conditioning.

**Strengths:**

1. The two-stage design effectively combines the stable training of Flow Matching and the efficient fine-grained generation of GAN, addressing the slow convergence/mode collapse of GANs and high computational cost of diffusion methods.
2. For audio’s unique properties, the authors propose endpoint estimation and spectral energy-adaptive loss scaling to improve Flow Matching, significantly enhancing generation quality in silent regions and perceptual consistency.
3. The multi-resolution network structure outperforms single-resolution designs in modeling audio complexity, providing a powerful backbone for generative learning.

**Weaknesses:**

1. Compared to BigVGAN-v2 trained on a larger dataset, it still has a slight gap in some metrics, suggesting limitations in generalization to larger-scale data.
2. The one-step model’s performance at low bandwidth (1.5 kbps) is inferior to its two-step version and some competitors, leaving room for improvement in low-bandwidth audio generation.

**Questions:**

1. What is the majoy difference between the proposed method and PeriodWave-Turbo? Is it just the improved Flow Matching model?
2. Can this improved Flow Matching strategy be applied to text to speech/audio tasks?

**Details Of Ethics Concerns:**

No concerns.

---

> ### Author Response · Authors · 2025-11-27
> **Thanks and response to concerns (part 1)**
>
> We sincerely thank the reviewer for the valuable and insightful comments. Below, we address each of the reviewer's concerns in detail.
>
> > Compared to BigVGAN-v2 trained on a larger dataset, it still has a slight gap in some metrics, suggesting limitations in generalization to larger-scale data.
>
> In response to Reviewer X1ZZ's suggestions, we have added TTS vocoder comparison results (Table 8) and 4-step Flow2GAN results. Since the TTS model generates log-Mel spectrograms, we have switched conditioning from Mel to log-Mel to better match it and for fair comparison. We have also updated Table 1 with log-Mel results (as it yields slight improvements), with Mel vs. log-Mel ablation results shown in Appendix Table 11.
>
> |Model | Condition | PESQ$\uparrow$ | ViSQOL$\uparrow$ | V/UV F1$\uparrow$ | Periodicity$\downarrow$ | FSD$\downarrow$ |
> |-|-|-|-|-|-|-|
> |Flow2GAN, 1-step | Mel | **4.288** | 4.939 | **0.976** | **0.058** | **0.027** |
> | Flow2GAN, 1-step | log-Mel | 4.189 | **4.957** | 0.975 | 0.063 | 0.028 |
> | Flow2GAN, 2-step | Mel | **4.454** | 4.966 | 0.981 | 0.045 | **0.019** |
> |  Flow2GAN, 2-step | log-Mel | 4.440 | **4.979** | **0.983** | **0.044** | 0.023 |
> | Flow2GAN, 4-step  | Mel | **4.484** | 4.966 | 0.984 | 0.039 | 0.017 |
> | Flow2GAN, 4-step  | log-Mel | **4.484** | **4.986** | **0.985** | **0.037** | **0.016** |
>
> **Now our 2-step and 4-step Flow2GAN models conditioned on log-Mel outperform BigVGAN-v2 across most metrics, including PESQ, ViSQOL, V/UV F1, and Periodicity.**
>
> | Model | PESQ$\uparrow$ | ViSQOL$\uparrow$ | V/UV F1$\uparrow$ | Periodicity$\downarrow$ | FSD$\downarrow$ | SMOS$\uparrow$ | MOS$\uparrow$ |
> |-|-|-|-|-|-|-|-|
> | BigVGAN-v2 | 4.379 | 4.971 | 0.978 | 0.055 | **0.014** | **4.65 $\pm$ 0.11** | **4.59 $\pm$ 0.10** |
> | Flow2GAN, 1-step | 4.189 | 4.957 | 0.975 | 0.063 | 0.028 | 4.44 $\pm$ 0.14 | 4.39 $\pm$ 0.15 |
> | Flow2GAN, 2-step | 4.440 | 4.979 | 0.983 | 0.044 | 0.023 | 4.53 $\pm$ 0.13 | 4.56 $\pm$ 0.11 |
> | Flow2GAN, 4-step | **4.484** | **4.986** | **0.985** | **0.037** | 0.016 | 4.60 $\pm$ 0.14 | 4.58 $\pm$ 0.14 |
>
> For TTS performance, all Flow2GAN models outperform BigVGAN-v2 in both SIM-o and UTMOS, with the 2-step and 4-step models achieving significantly better UTMOS:
>
> |Model |  Mel-type | WER(%)$\downarrow$ | SIM-o$\uparrow$ | UTMOS$\uparrow$ |
> |-|-|-|-|-|
> |BigVGAN-v2 | librosa | **1.82** | 0.66 | 3.70 |
> | Flow2GAN, 1-step | torchaudio | 1.94 | **0.67** | 3.75 |
> | Flow2GAN, 2-step | torchaudio | 1.91 | **0.67** | 4.11 |
> | Flow2GAN, 4-step | torchaudio | 1.89 | **0.67** | 4.18 |
> | Flow2GAN, 1-step$^{\dagger}$ | torchaudio | 1.88 | **0.67** | 3.83 |
> | Flow2GAN, 2-step$^{\dagger}$ | torchaudio | 1.88 | **0.67** | 4.21 |
> | Flow2GAN, 4-step$^{\dagger}$ | torchaudio | 1.87 | **0.67** | **4.26** |
> Note: $^{\dagger}$ denotes adding noise to log-Mel during GAN fine-tuning.
>
> > The one-step model’s performance at low bandwidth (1.5 kbps) is inferior to its two-step version and some competitors, leaving room for improvement in low-bandwidth audio generation.
>
> Yes, at 1.5 kbps bandwidth, compared to some competitors, the 1-step Flow2GAN achieves lower SMOS and MOS, but achieves notably better PESQ, ViSQOL, and FSD.
> | Bandwidth (kbps) | Model | PESQ$\uparrow$ |ViSQOL$\uparrow$ | FSD $\downarrow$ | SMOS$\uparrow$ | MOS$\uparrow$ |
> |-|-|-|-|-|-|-|
> | 1.5 | EnCodec | 1.368 | 3.409 | 1.996 | 1.69 $\pm$ 0.14 | 2.09 $\pm$ 0.27 |
> | 1.5 | MBD | 1.457 | 3.030 | 7.734 | 2.67 $\pm$ 0.18 | 2.83 $\pm$ 0.28 |
> | 1.5 | RFWave (10-step) | 1.600 | 3.272 | 2.986 | 2.87 $\pm$ 0.23 | 2.91 $\pm$ 0.23 |
> | 1.5 | PeriodWave-Turbo (4-step) | 1.260 | 3.308 | 4.055 | 1.55 $\pm$ 0.16 | 1.47 $\pm$ 0.19|
> | 1.5 | Flow2GAN, 1-step | 1.739 | 3.582 | 1.210 | 2.43 $\pm$ 0.20 | 2.83 $\pm$ 0.21 |
> | 1.5 | Flow2GAN, 2-step | 1.803 | 3.609 | 1.152 | 3.04 $\pm$ 0.20 | **3.86 $\pm$ 0.14** |
> | 1.5 | Flow2GAN, 4-step | **1.925** | **3.662** | **1.069**| **3.17 $\pm$ 0.19** | 3.40 $\pm$ 0.18 |

---

> ### Author Response · Authors · 2025-11-27
> **Thanks and response to concerns (part 2)**
>
> > What is the majoy difference between the proposed method and PeriodWave-Turbo? Is it just the improved Flow Matching model?
>
> Our method shares a similar high-level idea with PeriodWave-Turbo but builds upon our **improved Flow Matching formulation specialized for audio modeling**, incorporating **endpoint estimation reformulation** and **spectral energy-adaptive loss scaling.** This improved Flow Matching provides stronger pretrained weights for GAN generator initialization. As shown in ablation Table 3, using standard Flow Matching for pre-training as in PeriodWave-Turbo results in significantly lower performance after GAN fine-tuning, especially the one-step, demonstrating the superiority of our improved Flow Matching objective. Notably, PeriodWave-Turbo does not explore one-step results.
>
> | Pretrain | GAN finetuned 1-step | GAN finetuned 2-step |
> | - | - | - |
> | Standard Flow Matching | PESQ=3.730,ViSQOL=4.853 | PESQ=4.257,ViSQOL=4.933 |
> | Our improved Flow Matching | PESQ=**4.303**,ViSQOL=**4.942** | PESQ=**4.471**,ViSQOL=**4.969** |
>
> Additionally, our proposed multi-resolution structure, which processes Fourier coefficients, enhances modeling capability compared to single-resolution designs, contributing to the overall superior quality-efficiency trade-offs.
>
> > Can this improved Flow Matching strategy be applied to text to speech/audio tasks?
>
> It would be interesting to investigate our improved Flow Matching strategy in mel-based TTS setups. Intuitively, the endpoint estimation reformulation and spectral energy-adaptive loss scaling should also benefit mel-spectrogram generation, which also contains silent regions. One could further explore directly predicting waveforms from texts or coarse semantic tokens with this improved Flow Matching strategy. Due to time constraints and we focus on audio generation in this work, we leave this for future work.

---

### Author Response · Authors · 2025-11-27
**Thanks for the reviews and summary of key paper revisions**

We would like to express our sincere gratitude to all the reviewers for their insightful and constructive comments. We received initial scores of 6, 6, 6, and 4. The reviewers appreciated our contributions: **the overall Flow2GAN framework that combines the strengths of Flow Matching and GAN-based generation** (Reviewer KvbH, AhmV, and Vxds); **the improved Flow Matching specially for audio modeling** through endpoint estimation reformulation (Reviewer KvbH, AhmV, Vxds, and X1ZZ) and spectral energy-adaptive loss scaling (Reviewer KvbH, Vxds, and X1ZZ); the **multi-resolution network architecture** (Reviewer KvbH and Vxds).

Below, we provide a summary of the revisions made in response to the reviewers' suggestions.
1) In response to Reviewer X1ZZ's suggestions, we have made the following improvements: **added TTS vocoder comparison results in Table 8, added 4-step Flow2GAN results, included the MOS metric in Tables 1 and 2, and added PeriodWave-Turbo's Encodec-based results in Table 2** using its released checkpoints from GitHub. Since the TTS model generates log-Mel spectrograms, we have switched conditioning from Mel to log-Mel to better match it and for fair comparison. Consequently, we have also **updated Table 1 with log-Mel results** (which yielded slight improvements) and included Mel vs. log-Mel ablation results in Appendix Table 11. For ablation studies, we maintained Mel conditioning. In addition, as we have added 4-step results, we have updated the terminology "one-step/two-step" to "few-step" in title, abstract and main paper.
2) In response to Reviewer AhmV's suggestion, we have **added comparation experiments with ShortCut models, a recent few-step Flow Matching variant, with results shown in Table 5**.
3) We have also improved clarity by: **distinguishing our spectral energy-adaptive loss scaling from the frame-level way used in RFWave**, in response to comments from Reviewer AhmV and X1ZZ; **clarifying that our GAN fine-tuning is built upon our audio-tailored Flow Matching, a key distinction from PeriodWave-Turbo**, in response to comments from Reviewer KvbH, Vxds, and X1ZZ.
4) We have adjusted the tables positions to optimize space utilization.

---

### Author Response · Authors · 2025-12-02
**Response summary for AC**

We thank the AC and reviewers for their careful evaluation and constructive comments. Our work, Flow2GAN, combines Flow Matching and GAN for high-fidelity audio generation in few sampling steps. We improve flow-matching for audio modeling by reformulating the objective as endpoint estimation and scaling the loss based on spectral energy, providing a strong initialization for GAN fine-tuning. In addition, our multi-resolution architecture improves modeling capabilities. Flow2GAN achieves superior quality-efficiency trade-offs over existing methods, e.g., comparable generation quality to BigVGAN-v2 but much faster. The reviewers appreciated these contributions, with initial scores of 6, 6, 6, and 4.

We have provided detailed responses to each reviewer, mainly including: 1) **added TTS results comparison using various vocoders, 4-step Flow2GAN results, MOS metric, and PeriodWave-Turbo's Encodec-based results** (Reviewer X1ZZ); (2) **added comparison with ShortCut models** (Reviewer AhmV); (3) **clearer clarifications distinguishing our spectral energy-adaptive loss scaling from the frame-level method in RFWave** (Reviewers AhmV, X1ZZ), and **emphasizing that our GAN fine-tuning builds on our audio-tailored Flow Matching, a key distinction from PeriodWave-Turbo** (Reviewers KvbH, Vxds, X1ZZ).

Unfortunately, we have not yet received reviewer feedbacks on our responses, but we are confident that our supplementary experiments and detailed clarifications comprehensively address all reviewer concerns. We welcome further discussions and are happy to address any additional questions if the discussion opens again. We hope the AC will carefully consider these responses when making the decision.

---

### Meta-Review · Area_Chair_kWHw · 2026-01-09

**Summary:**

This paper proposes Flow2GAN, a two-stage framework for few-step high-fidelity audio generation. The approach first performs Flow Matching pretraining using an endpoint-prediction reformulation together with a spectral energy-adaptive loss scaling, and then applies lightweight GAN fine-tuning to obtain 1/2/4-step generators. A multi-resolution Fourier/iSTFT backbone is used to improve modeling across time-frequency scales. Overall, the work is empirically strong and clearly targets an important practical objective in audio generation: achieving favorable quality--efficiency trade-offs at very low sampling step counts.

**Reviewer Concerns:**

Reviewers generally appreciated the motivation and the reported performance, but raised concerns about novelty and positioning relative to closely related prior work, especially conditional flow matching vocoders (e.g., PeriodWave/RFWave), energy/variance-based reweighting schemes, and FM+GAN post-training approaches for acceleration (e.g., PeriodWave-Turbo).
Reviewer X1ZZ in particular emphasized that endpoint prediction and energy-based scaling resemble existing techniques, and that the manuscript initially did not sufficiently differentiate its contributions from prior systems, especially regarding the GAN fine-tuning stage.
Additional concerns included missing comparisons to the most relevant few-step competitors (e.g., shortcut-style approaches), and requests for broader perceptual evaluation beyond purely spectral-domain analysis.

The rebuttal and revised version substantially strengthen the empirical and comparative basis.
The authors add MOS evaluation, report 4-step results, and include a two-stage TTS vocoder comparison to reduce the risk of overfitting to ground-truth conditioning.
They also add direct comparisons requested by reviewers, including ShortCut models and PeriodWave-Turbo in the EnCodec-token setting using released checkpoints.
Methodological clarity is improved by explicitly distinguishing the proposed time--frequency-aware loss scaling from per-frame energy-balanced losses, clarifying the multi-branch iSTFT aggregation detail, and calibrating claims by removing unsupported statements regarding GAN mode collapse mitigation. These changes address most concrete reviewer requests and reduce ambiguity in the experimental story.

The primary remaining issue is that the incremental nature of several ingredients can still be debated.
While the rebuttal provides evidence that the audio-tailored Flow Matching objective yields materially stronger initialization for GAN fine-tuning (notably improving one-step performance relative to standard Flow Matching pretraining), the paper’s contribution is best viewed as a well-validated and carefully assembled system improvement rather than a fundamentally new modeling paradigm.
In addition, the positioning should remain cautious and emphasize quality--efficiency trade-offs rather than broad claims of universal state-of-the-art across all speed operating points, particularly given the existence of extremely fast vocoders that occupy different points on the Pareto frontier.

**Reviewer Scores:**

Initial scores were 6, 6, 6, and 4.
The rebuttal additions directly address the most substantive experimental requests, and are likely to improve overall confidence in the empirical claims.
While the highest-confidence novelty critique (X1ZZ) may still view the contribution as incremental, the clarified distinctions and stronger comparisons plausibly move the overall assessment to borderline acceptance, provided the final camera-ready further calibrates novelty and claims.
\begin{itemize}
  \item KvbH (initial 6) likely remains 6.
  \item AhmV (initial 6) likely remains 6, possibly raise score if fully satisfied by the added comparisons and clarifications.
  \item Vxds (initial 6) likely remains 6.
  \item X1ZZ (initial 4, high confidence) might increase to 5 as the requested experiments being added, but novelty concerns likely persist.
\end{itemize}
However, the low confidence of reviewer Vxds(3) and KvbH(2) shows that the paper is not universally convincing, and the novelty debate remains contentious.
The paper provides a strong empirical contribution and a practical recipe that achieves compelling few-step quality-efficiency trade-offs, and the rebuttal addresses the main experimental and comparison gaps raised by reviewers.
Acceptance is contingent on careful final positioning of the contributions as an empirically validated hybridization and on maintaining clear, conservative claims about novelty and scope.

---

### Decision · Program_Chairs · 2026-01-26

Accept (Poster)